# Potential Applications of Core-Shell Nanoparticles in Construction Industry Revisited

**Ghasan Fahim Huseien**

Department of the Built Environment, College of Design and Engineering, National University of Singapore, Lower Kent Ridge, Singapore 117566, Singapore; bdggfh@nus.edu.sg; Tel.: +65-83057143

**Abstract:** The demand of high performance and environmentally sustainable construction materials is ever-increasing in the construction industry worldwide. The rapid growth of nanotechnology and diverse nanomaterials' accessibility has provided an impulse for the uses of smart construction components like nano-alumina, nano-silica, nano-kaolin, nano-titanium, and so forth Amongst various nanostructures, the core-shell nanoparticles (NPs) have received much interests for wide applications in the field of phase change materials, energy storage, high performance pigments, coating agents, self-cleaning and self-healing systems, etc., due to their distinct properties. Through the fine-tuning of the shells and cores of NP$_S$, various types of functional materials with tailored properties can be achieved, indicating their great potential for the construction applications. In this perception, this paper overviewed the past, present and future of core-shell NPs-based materials that are viable for the construction sectors. In addition, several other applications of the core-shell NPs in the construction industries are emphasized and discussed. Considerable benefits of the core-shell NPs for pigments, phase change components, polymer composites, and self-cleaning glasses with enhanced properties are also underlined. Effect of high performance core-shell NPs type, size and content on the construction materials sustainability are highlighted.

**Keywords:** nanoparticles; core-shell materials; pigments; polymer; phase change materials

## 1. Introduction

Nanotechnology is defined as the manipulation of shape and structure of materials at the nanoscale that can be used to design, characterize and produce valuable structures, devices, and systems. The nanoscale refers to the objects with sizes between 1 and 100 nm in dimensions (1 nm = $1 \times 10^{-9}$ m). Despite many challenges in manipulating the engineering materials at such a small scale, the recent advancements of various imaging techniques made it possible to design, manufacture, and study their behaviours at the nanoscale. Amongst all the nanoscale structures produced by the top-down or bottom-up approaches, the nanoparticles (NPs) became most interesting. These are usually produced in the form of very fine powders or colloidal suspensions [1–4]. Various emerging properties of these NPs mainly depend on their individual components that are appreciably different from their bulk counterparts [5,6]. The NPs are unique because of their enlarged surface area, quantum size effects, improved absorbance, uniformity, and surface functionalization. The quantum size effect of the NPs is responsible for their distinct physicochemical characteristics useful for sundry applications [3,7–10].

Selected as one of the ten topmost targeted applications of nanotechnology to ameliorate some of the most significant issues in the developing nations, construction and architecture industries stand to be substantially enhanced by the uses of nanomaterials [11,12]. Despite their ongoing uses within these contexts [11,13,14], the future of nanotechnology in these industries is predicted to further increase the application feasibilities. Among these expected outcomes, improvements in the building material' properties by making them stronger, durable, and lighter is the main focus [15–17]. These enhancements are brought

by introducing novel collateral functions like self-heating, anti-fogging, and energy-saving coatings, and so on [18–21]. In addition, the key components for the maintenance of instruments such as sensors that detect and report structural health have been developed to gain more benefits of these nonmaterials [22]. Despite various advantages of these new technologies, an emphasis should be placed on the risk-assessment of their intended uses, wherein the fallout can be severe. One such recent example is the deliberate and widespread use of supposedly beneficial chemical dichlorodiphenyltrichloroethane (DDT) that was released to control malaria and various water-borne diseases. However, instead proved to be carcinogenic to humans it became toxic to numerous bird species, and hazardous to environment [23]. This illustrates the importance of a proactive and meticulous approach for the risk assessment of new technologies, without which, devastating impacts to ecosystems and human health cannot be prevented.

Buildings have remarkable rate of power consumption at 45% of global energy [24,25]. Many passive cooling methods have been used and in addition, phase change materials (PCM) are installed within these buildings for the purpose of promoting temperature moderation, stopping heat from accumulating, improved heat absorption and minimize indoor heat gain. The method in which PCM stores thermal energy is effective in improving the buildings' aggregate heat capacity. Interest has been strong in PCMs that has high energy density to be deployed in buildings with high thermal inertia in order to save a high amount of energy. However, PCMs have their own drawbacks and the primary one being extra time required to charge/discharge energy process as well as storage performance, which happens due to poor thermal conductivity. Therefore, attention is focused on improving its thermal conductivity through the use of nanotechnology and nanomaterials. There has been a rapid development lately within the nanomaterials field resulting in the latest technology with Nano-sized particles in improving the PCM's thermophysical properties. PCMs have several thermal and physical qualities such as viscosity, heat capacity, super-cooling and thermal conductivities. These attributes could be significantly improved through dispersal of thermal conductive nanoparticles including nanometal-oxide, nanocarbon and nanometals. The technologies of core-shell and nanoparticles are widely adopted to improve the materials properties and thermal performance, which is appropriate in passive-cooling within the built-environment.

In the construction industry, one of the possible solutions for a sustainable future is to introduce novel technologies to improve the durability of materials and increase the life span. Presently, nanotechnology creates new possibilities to control and improve material properties for civil infrastructures. By combining various engineering, chemical, and biological approaches, the nanotechnology can be used for the sub-atomic manipulation of materials. To synthesize NPs, diverse chemical, biological, physical, and even hybrid techniques can be used. In this regard, this review discusses and explains the role of nanoscience and nanotechnology in the development of potential core-shell NPs applicable in the construction industry (Figure 1). Also, diverse potential applications of core-shell NPs -based high performance construction materials rooted from the state-of-the-art research are emphasized.

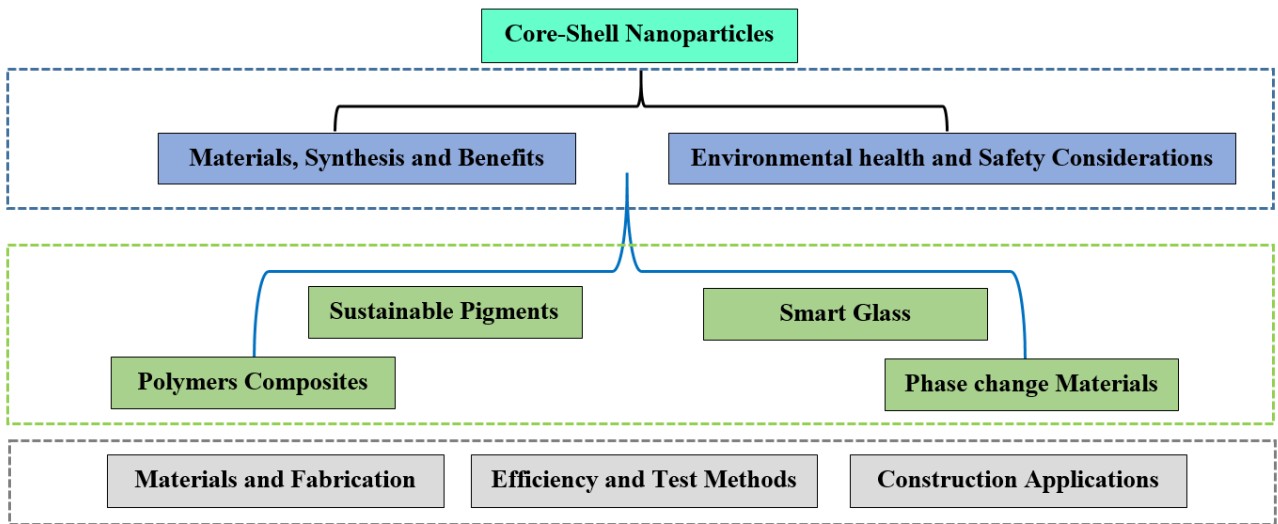

**Figure 1.** Flow chart of core-shell nanoparticles, synthesis, efficiency and construction applications.

## 2. Core-Shell NPs Synthesis and Benefits

Nanotechnology encompasses various methods of synthesis (biological, engineering, chemical and hybrid) to customize the atomic-scale properties of materials. To produce the core-shell NPs both top-down and bottom-up approaches are routinely utilized. Top-down approach incorporates the conventional workshops with microfabrication tools in addition to the equipment that are externally controlled that are used to mill, cut, shape and mould the materials accordingly to the requirement [26,27]. The lithographic and mechanical techniques are the conventional top-down approach. The lithographic techniques involve the use of electron or ion beam, UV, scan probing, optical near field scanning and laser-beam processing. In addition, the mechanical techniques involve the machines that grind, cut and polish the materials according to the required specifications [28–31]. Conversely, the bottom-up technique is used to assemble materials in the desired form from their chemical composition down to the molecular level. Examples of typical bottom-up technique include chemical vapour deposition, laser-induced assembly, chemical synthesis, self-assembly, colloidal aggregation as well as film deposition and growth [32,33].

Both approaches have many advantages and disadvantages. However, the main advantage of the bottom-up approach is related to its cost-effectiveness that can fabricate significantly smaller particles than the top-down approach. This is because of its precision as the product is produced by assembling it down to molecular level. Thus, it is possible to have total control and almost no energy loss in the entire production process. The preparation of core/shell NPs necessitates total control in order to coat the shell materials uniformly as the particles are formed. Therefore, bottom-up approach is more suitable for such synthesis. Hybrid approach involves the use of both of the aforementioned techniques. For instance, the core particles can be produced via the top-down approach. Conversely, the bottom-up approach can address the uniformity of the shell thickness. It is recommended to apply micro-emulsion for an accurate size and thickness regulation of the shell because water droplets can act as nano-reactors. More researchers have been focusing on the core-shell NPs due to their suitability to be used extensively in diverse fields such as electronics, optics, chemistry, biomedicine, medicines and catalysis, etc.

The core-shell NPs have high functioning and distinct properties such as different materials can be used for the core or shell. The core or shell can be highly customizable by modifying the properties through controlling the materials or the core to shell ratio [34]. Also, it is possible to modify the core particles' reactivity and thermal stability via the adjustments to the shell coating material, leading to improved stability and dispersion of the core particles. This indicated that each particle can possess exclusive properties depending on the materials being used during the fabrication. Such technique is renowned

because through the application of appropriate materials it can customize the surface function according to the environment [35]. The benefits of coating core particles include improved function, surface modifications, stability, dispersion, core release control and significant decrease in the use of precious material. The core-shell particles, as the name suggests, contain a shell and a core wherein a shell can be produced by the same or different materials used for the core [36–38]. Figure 2 shows various core-shell particles where the colours are used to differentiate between them wherein a core can consist of a single sphere (Figure 2a) or multiple spheres that are smaller in size (Figure 2b). Furthermore, a shell may be hollow with one small sphere inside, resembling the yolk-shell structure (Figure 2c) [39].

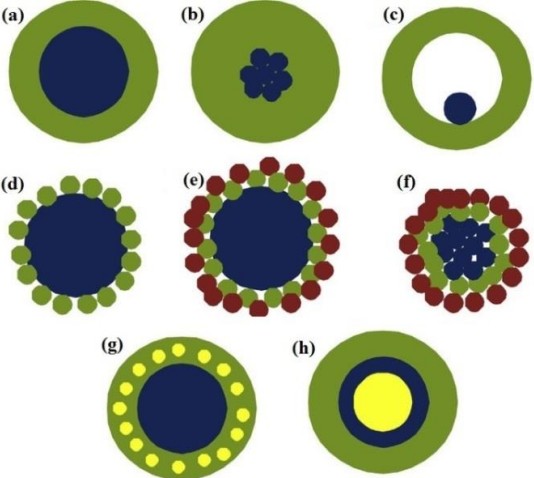

**Figure 2.** Schematic representation of different types of core–shell particles (**a**) single sphere (**b**) multiple spheres are smaller in size) (**c**) yolk shell (**d**) large core sphere with one layer of many smaller spheres (**e**) large core sphere with two layers of many smaller spheres (**f**) simply a collection of core spheres (**g**) smaller spheres into the shell (**h**) multiple shells [36].

Figure 2 shows three forms of shell structure like a continuous layer (Figure 2a–c), a larger core sphere that contains many smaller spheres (Figure 2d,e) or simply a collection of core spheres (Figure 2f) [40]. The intricacy of the core-shell structure can be manipulated by inserting smaller spheres into the shell (Figure 2g) [41] that can also be done through multiple shells (Figure 2h) [42,43]. The core-shell NPs can be made using the physical or chemical approach including the chemical deposition, physical vapour and wet chemistry. Generally, the synthesis of core-shell particles involves different stages. First, the core particles are synthesized followed by the formation of shell onto the core particle. This method depends on the type of core and shell materials [41]. The main purpose of producing the core-shell particles is to achieve suitable unconventional novel materials and structures. Consequently, the materials with desirable attributes such as active particles with high stability, biocompatibility and synergy effect can be achieved [44].

Diverse industries utilise a basic nanomaterial to synthesize core-shell NPs. The speed, simplicity, environmental friendliness and cost effectiveness of the method as well as products are prerequisites for the synthesis of these core-shell structures. Many methods have been established to meet the aforementioned requirements. These include the electrochemical dealloying, sol-gel process, sonochemical process, microwave synthesis, multi-step reduction, microe-mulsion, epitaxial growth, and Stöber method. Hybrid method involves the unification of more than one of the aforementioned methods. Generally, sol-gel process is widely used to produce the core-shell NPs. Sol-gel method for the synthesis of core-shell NPs offers an additional control during the reaction process of solid materials. Homogenous multi-component systems especially mixed oxides can easily be produced through the mixing of solutions containing molecular precursors. Essentially, this method yields solid materials (small molecular clusters), especially metal oxides like $SiO_2$ and $TiO_2$. Preparation of metal oxides using sol-gel process involves the conversion of monomer

into a colloidal solution (sol). The solution is the precursor to be used for combination of network including discrete particles or network polymers. Usually, various metal alkoxides are used as precursor. Sol is produced when a chemical reaction occurs and eventually become a diphasic substance that has property similar to gel, implying both liquid and solid. The morphology of these phases may either be discrete particles or continuous polymer networks. Turning the colloids into the properties like gel necessitate the removal of large volume of liquids in the events from the volume of particle density that is significantly low. One of the simplest ways to achieve this is to wait for an adequate time for the sedimentation to occur before disposing the remaining liquid. In addition, the centrifugation can be applied to accelerate the phase separation. Sol-gel is a more common wet-chemical method that is used to synthesise core-shell NPs [45–47].

Microemulsions are a mixture of isotropic liquid composed of surfactant, oil, water and more commonly co-surfactant. It has a clear appearance and a stable thermodynamic system in the presence of salt and other ingredients in the liquid form. The oily substance may be due to complex mixtures of various types of hydrocarbons. In comparison to conventional emulsions, microemulsions are synthesised through different mixing components and do not need high shear conditions during the production process. The microemulsions are categorized as direct (dispersion of oil in water, o/w), reversed (dispersion of water in oil, w/o) and bycontinuous types. These microemulsions belong to the ternary systems wherein two immiscible substances (water and oil) which forms separate layers co-exists with a surfactant, resulting in a monolayer that form between the immiscible substances from the surfactant's molecules. In the oil phase, the hydrophobic tails of the surfactant molecules would dissolve. However, in the liquid phase, the hydrophilic head groups would dissolve. Two-step microwave irradiation is the conventional method for rapid synthesis of gold and silver core-shell bimetallic NPs. In this technique, a bilayer organic barrier is developed surrounding the core. The desired capping agents are the citrate and ascorbic acid that facilitates the formation of core and shell material, developing a well-defined boundary layer. The boundary layer is significant for the synthesis process of various core-shell particles that are ultimately used to create the customised bimetallic core-shell NPs of desired morphology wherein the cores are triangular or spherical in shape.

The high-pressure chemical vapour deposition method is an alternative in producing the core-shell materials including nanotubes. Nikolaev et al. [48] founded this method to produce the single-walled carbon nanotube (SWCNTs). In this process, a small amount of $Fe(CO)_5$ is used to comb CO. Then, the mixture is passed to a heat reactor. El-Gendy et al. [49] used this technique to make NPs coated with various materials like Fe, Co, Ni, FeRu, CoRu, NiRu, NiPt, and CoPt. In this method, the reactor's temperature and pressure can be accurately controlled to tailor the core-shell NPs properties needed for the specific applications. Earlier, the metal-organic precursors called the metallocenes or metals that are rich in carbon were used. These precursors were inputted into a thermostatic sublimation chamber before releasing argon gas for pushing the vapour into the hot zone of the chamber. First, the precursor broke down the NPs within the cooling finger before turning into the gas phase within the hot zone for the supersaturation. Upon the initiation of the supersaturation process, the NPs were nucleated. Furthermore, careful adjustments can be made to the temperature and pressure/temperature within the corresponding sublimation chambers and chemical vapour deposition reactor in order to control the desired degree of supersaturation. At high pressure, the collision probability of gas atoms increases, thus reducing the rate of atoms diffusion from the original location. It is worth noting that when the diffusion rate is poor, the supersaturation does not occur. In this situation, the cooling finger contains the deposits of tiny clusters of atoms or individual atoms.

$Fe_2O_3$ with graphene shells as coating was prepared using the wet chemical technique [50]. Oleic acid and 1-octadecene were mixed in a solution before being placed in the reflux reactor and heated to 320 °C to dissolve the iron oleate. Next, the solution was washed with ethanol and acetone to obtain the iron oxide particles. Normally, the Stöber process involves the preparation of $SiO_2$ particles [51] with total control of uniformity in

size [52]. These particles offer numerous applications in the field of materials science and engineering. Since the discovery of this method by Werner Stöber et al. [51], it remains the most renowned wet chemistry approach for the NPs synthesis [53]. Being a sol-gel process, the chemical tetraethyl orthosilicate (TEOS) act as the precursor immersed in water. Alcoholic solution is added to form a reaction, forming new molecules that agglomerate to create larger clusters. Du et al. [54] used the sol-gel approach to make $SiO_2$ shell as a coating agent for the $Fe_3O_4$ NPs, eventually producing the core-shell structure. In this two-step procedure, the co-precipitation was first initiated to obtain $Fe_3O_4$ NPs. Next, it caused a reaction with tetramethyl ammonium hydroxide (TMAOH), forming a liquid solution that contained the proposed particles. In the second stage, $SiO_2$ was produced through the hydrolyzation of TEOS in order to limit the formation of $Fe_3O_4$.

Figure 3 shows the sol-gel unified annealing approach used by Li et al. [55] to produce $ZnSiO_3$/ZnO core-shell NPs. In this experiment, the reason for combining these two methods was to produce broad band-gap core-shell NPs of Zinc Silicate-Zinc Oxide ($Zn_2SiO_4$@ZnO). First, the reaction between $Na_2SiO_3$/$ZnCl_2$ was initiated to form $ZnSiO_3$, which in turn produced the shells with varied thickness before being used to coat ZnO NPs. A low annealing temperature of 780 °C was set. Finally, the reaction between amorphous $ZnSiO_3$ and ZnO occurred, forming a crystalline $Zn_2SiO_4$ shell. Chai et al. [45] adopted this technique to make core-shell $Fe_3O_4$@$SiO_2$ NPs. The first step was to fabricate $Fe_3O_4$ NPs via the solvothermal technique. Next, the hydrolyzation of tetraethyl orthosilicate resulted in $SiO_2$ that acted as the coating agent for $Fe_3O_4$ NPs.

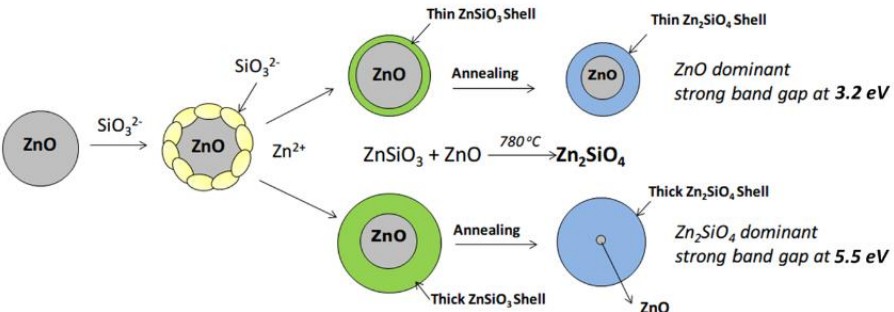

**Figure 3.** Core-shell particles synthesis using sol-gel combined annealing method [55]. Reproduced with permission from Li, Z, et al., Materials Chemistry and Physics; published by Elsevier, 2020.

A two-step reduction technique was also used [56] to make epitaxial Au@Ni core-shell nanocrystals. In this process, various materials such as decahedral, octahedral, triangular and hexagonal plate-like as well as icosahedral were mixed initially. Subsequently, ethylene glycol (EG) was used for the reduction of $HAuCl_4$ before being placed in a microwave with polyvinylpyrrolidone (PVP) that acted as a polymer surfactant to be heated. The core seeds were produced at this stage and subsequently the oil bath was heated to reduce $Ni(NO_3)_2 \cdot 6H_2O$ in EG in the presence of NaOH and PVP. Eventually, the Ni shells were overgrown within the Au core seeds. Fan et al. [57] used similar technique but focused on the seed-mediated growth. Herein, Au cores were made in the liquid form to achieve bimetallic core-shell nanocubes. Comprehensive assessment was made upon the heterogeneous core-shell formation on the four common metals like gold, silver, palladium and platinum. This experiment constituted the following bases: (a) the general conditions and growth modes to attain conformal epitaxial structures and (b) heterogeneous nucleation and formation of various noble metals. In addition, three types of growth modes for the gold cores with heterogeneous metal shells were identified: conformal epitaxial growth (Au@Pd and Au@Ag nanocubes), island growth (Au@Pt nanospheres) and heterogeneous nucleation. Further findings include two metals with comparable lattice constants where the mismatch was less than 5%. These findings were consistent with other studies (Au@Ag (lattice mismatch, 0.2%), Au@Pd (4.7%), and Pt@Pd (0.85%)) [54–56].

Tsuji et al. [58] used one-polyol technique to make Ag@Cu core-shell NPs with a high yield. The method involved the use of bubbling Ar gas with added reagents like AgNO₃ and Cu(OAc)₂ H₂O. This two-step process was used to synthesize Ag@Cu particles through AgNO₃ reduction in EG. The Cu shells were developed by separating the Ag cores from AgNO₃, before Cu(OAc)₂.H₂O was added. This procedure failed because no Cu@Ag core-shell particles were nucleated instead the Cu/Ag bi-compartmental particles were appeared. Later, various experimental processes were combined at different reaction temperatures and heating times to produce Ag@Cu particles. It was found that the optimal condition for producing Ag@Cu particles is to add two reagents in reverse. At the beginning of the process, 8 mL of 15.9 mM Cu (OAc)₂.H₂O was added in EG plus 8 mL of 477 mM poly(vinylpyrrolidone) (PVP, MW: 55,000 monomer units). A 100-mL three necked flask was used for the solution mixing. Ar was bubbled for 10 min at room temperature to completely remove oxygen from the solution followed by soaking in an oil bath at a temperature of 180 °C. The solution continued to bubble while the temperature was raised to 175 °C. Afterwards, the reagent solution was added with 2 mL of 15.7 mM AgNO₃ and left for 20 min at 175 °C. Finally, 7.0 mM, 212 mM and 1.7 mM of Cu (OAc)₂.H₂O, AgNO₃ and PVP, respectively. Further investigation was conducted by varying the reaction time on the reagent solution to determine the growth mechanism of Ag@Cu.

Chae et al. [45] produced Fe₃O₄@SiO₂ by the customised Stober method. The solution of 4 g Fe₃O₄ particles was ultrasonicated and extra tetraethyl orthosilicate was added to raise the volume from 4 to 40 mL. A stable emulsion was obtained and it was further inserted into a mixture containing 50 mL of ethanol and 12 mL of NH₃ H₂O. The reaction solution was stirred at 400 rpm at room temperature for 4 h until the core-shell structured Fe₃O₄@SiO₂ NPs were separated using centrifugation. Figure 4 shows the entire process of Fe₃O₄@SiO₂ synthesis.

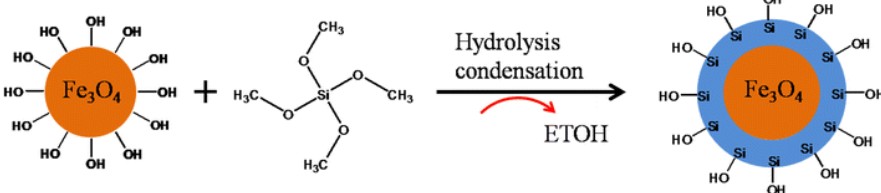

**Figure 4.** Stober method for Fe₃O₄@SiO₂ nanoparticles synthesis [45]. Reproduced with permission from Chae, H.S., et al., Colloid and Polymer Science; published by Elsevier, 2016.

Sharma et al. [59] conducted a similar experiment and demonstrated that it is possible to fabricate core-shell particles through the precipitation without the need of any surfactant. The outcome (the concentrations of the core-shell particles) was compared with those obtained using different anionic and non-ionic surfactants. The nano -TiO₂ was developed in the form of shell using fly ash. The surfactants were mainly used to strengthen the adhesion of the nano-titania shells to fly ash core. Yet again, different types of surfactants were used to test the strength of the TiO₂ adhesion onto fly ash. Another test was conducted without surfactant. When anionic surfactant was used, the resulting particles formed had remarkable pigment properties and reflectance in the near-infrared region, indicating their suitability towards cool coating applications. A solution of 70% ethanol was added in the sequence of fly ash, anionic (SDS) or non-ionic surfactant (TX-100) and finally titanium isopropoxide. Finally, the solution was stirred for two hours before being dried at 50–600 °C to achieve a powder.

Zhang et al. [60] made a study to produce PUA hybrid emulsion PA/PU with a ratio of 20 to 80 using semi-batch emulsion. In the experimental setup a digital thermometer, 250 mL four-neck glass flask containing a reflux condenser, mechanical stirrer and nitrogen gas inlet were used. The pre-emulsion was prepared by dissolving 2.0 g per 100 g of acrylic and PU content into the water before gradually adding 5.0 g of MMA, 5.0 g of BA and 0.015 g of AA (0.1 5 wt% of the overall MMA and BA weight). The solution was then stirred

before mixing additional 0.5 g. The main objective was to obtain 111.3 g of PU emulsion dispersion and 10% monomers from the reactor vessel. The temperature was set at 80 °C while the contents were stirred. Next, 0.4 g of KPS per 100 g acrylic monomers composed of 10% was added and continuously stirred for 30 min. Subsequently, the temperature was increased by 5 °C and simultaneously the leftover monomer pre-emulsion and initiator solution was flown into the task for 4 h at a constant flow rate. Next, the solution was left at 85 °C for 0.5 h with stirring and waiting for the temperature to drop. Lastly, the pH value was maintained at the desirable range after adding $NaHCO_3$.

## 3. Core-Shell NPs Based Sustainable Pigments

In the last decade, synthetic-coloured pigments have been launched in the market that resulted in more extensive scientific research focused on this area. Typical applications of these pigments are varnishes, paints, plastics and textiles, printing inks, building materials and rubber, ceramic glazes and leather decoration [61–63]. The definition of the pigment durability is connected to its ability of resisting weathering processes and negating deteriorating when being placed in an external environment [64]. Recent studies have shown that efficient energy consumption and environmental protection measures are deemed significant [65]. To address this issue, the production of both sustainable and durable pigments has become the fundamental requirement within the construction industry. Myriad of methods have been applied in order to increase the pigments' durability, and the most significant is known as the core-shell method [3,66–68]. There has been a surge of development of various chemical synthesis techniques in recent years. Such research has found that multi-component materials possess diverse compositions and structures. These attributes signify remarkable property type and they are applicable in many different types of fields [69–72]. There is even more research being conducted on their distinctive core-shell structure.

There are many advantages of the core-shell structure compared to other types of composite materials. One such advantage is their ability to generate or increase the strength of new chemical and physical capabilities, enabling maintenance on structural integrity, deter the core from breaking up to large particles and ascertaining dispersion effectively. In addition, they also provide conventional multi-functional compositions and structure with other advantages. Moreover, a synergetic effect between the shells and cores would even extend the performance further [73]. Science and technology field have been attentive on the phenomena of materials that are derived from the core-shell properties because they can be finely customised [61,74,75]. A shell domain cloaks a core structural domain within each of the core or shell particle. Materials that possess core or shell particles include inorganic solids, metals and polymers. There is no difficulty in modifying characteristics such as size and structures as well as the particles' composition in order to further customise their properties such as optical, magnetic, mechanical, thermal, electrical, catalytic and electro-optical attributes.

Core or shell morphology can be applied to produce hollow spheres and minimize the costs of precious materials. Thus, the materials with the reduced core costs can be coated to precious materials [76,77]. Particles with the size of less than 0.1 μm is classified as NPs and have been garnering much attention in research within the past few years. Essentially, NPs are smart materials with exclusive properties. Applications using NPs have more advantages compared to materials that have larger surface to volume ratio such as microscale, macroscale and bulk materials [78,79]. Due to the increased research on the NPs development, it is now possible to make NPs in symmetrical shape, such as spherical as well as other shapes including prism, hexagon, cube, wire, tube and rod [80–82]. Despite this achievement, the bulk of the research is still at early stage in terms of exploring the possible shapes that can be synthesised. There has been research that recently found the ease of production method for NPs that are non-spherical [83–85]. However, it must be stressed that NPs' properties are dependent on the actual shape and size. Such properties that are dependent on particle size include temperature barrier, magnetic saturation and

permanent magnetisation. Furthermore, coactivity of the nanocrystals is dependent on the shape of the particle as it has a direct influence on the surface anisotropy [86].

Rapid advancements are made in nanotechnology resulting in the founding of core-shell NPs, which is a leading functional material. This has attracted even more research conducted on various functional compositions core-shell NPs where it could be applied in many types of areas such as optics, catalysis, biomedicine, electronics and medicines [87]. Core-shell NPs possess beneficial physiochemical properties that are exclusive, and this attribute has garnered a lot of researchers' attention. The primary advantages of core-shell NPs are that it could increase protection level, encapsulation and controlled release [82,88]. The discovery of a variety of core/shell NPs leads to its applications to a variety of situations. However, the difficulty is to identify the individual type core/shell NPs that are applicable to the respective industries due to their multitude of types. Numerous studies on the core-shell NPs pigments are focusing on core/shell materials, production methods, distinctive properties and their applications. Herein, the main features of the core-shell NPs including their fabrication methods, inorganic materials and typical applications are emphasized. A discussion on diverse methods of production along with the classifications of the core-shell materials that are already being in use are outlined. The new fabrication methods of the core-shell NPs pigments within all research fields are emphasized. Finally, the application potential of core-shell NPs within paints designed for roads and other construction sectors are underscored.

### 3.1. Materials Based Shell Part

Several materials such as metals and biomolecules are used to create core-shell NPs. There are two components, the central core and an alternative core, which is the shell. The attributes of the core-shell nanostructures include their high thermal and chemical stabilities, low toxicity, high levels of solubility and high level of permeability for specifically targeted cell. Such properties enable them to have a vast potential for functional applications in many sectors. Furthermore, micro-nano scale core-shell particles have attributes that are exclusive and unique to them compared to other particles. Essentially, the attributes combined the materials' properties that are used for core and shell together along with smart properties that are formed via their materials. For the past few years, there have been an increased research interest in core-shell structures production [89]. This is particularly true within the pigment industry due to the high range of applications of core-shell materials in order to increase pigments' durability. The core-shell materials could be made of both organic and inorganic materials. For instance, Cao et al. [90] developed hybrid pigments which consist of inorganic-organic structure using a mixture of precipitated $SiO_2$ and $TiO_2$. In addition, dye core@silica shell structure was fabricated using the mesoporous soft template synthesis approach [91]. This section below explores the possibility of using inorganic materials to produce core-shells materials with the focus on $SiO_2$ and $TiO_2$.

### 3.2. Efficiency and Test Methods

Generally, the obtained core-shell NPs are characterized using diverse analytical methods such as SEM, LC-MS, XPS, FTIR, XRD, TEM, BET, Ultraviolet-visible Spectroscopy, Raman spectrum as well as Near-Infrared Reflectance and Photoluminescence Spectroscopy [37,38,47,92–96]. For instance, assessment of morphology, chromaticity and the structure of $\alpha$-$Fe_2O_3$@$SiO_2$ fabricated pigments can be tested by SEM, TEM, FTIR, XPS and XRD [88]. Figure 5a shows the XRD patterns of the pigments made of $\alpha$-$Fe_2O_3$@$SiO_2$ NPs, -$Fe_2O_3$@$SiO_2$ and $\alpha$-$Fe_2O_3$. The formation of the core-shell structures results in the diffraction peak of $\alpha$-$Fe_2O_3$@$SiO_2$ particles appeared in the 2θ range of 15°–25°, indicating the presence of amorphous $SiO_2$. Further calcinations at 1000 °C could change the diffraction peak to around 22°. The results of reddish colour pigment indicated that the amorphous shell has entered into a cristobalite phase. In addition, the formation of the core-shell structure weakened the $\alpha$-$Fe_2O_3$ diffraction peak. Figure 5b shows the FTIR spectrum of the reddish pigments of $\alpha$-$Fe_2O_3$, $\alpha$-$Fe_2O_3$@$SiO_2$ NPs, and $Fe_2O_3$@$SiO_2$. The

hydroxyl (–OH) stretching vibration bands were probed at 3423.50, 1627.85 cm$^{-1}$, 536.19 and 466.75 cm$^{-1}$, indicating a correlation to the O–Fe–O bands of α-Fe$_2$O$_3$. The band at 1091.66 and 470 cm$^{-1}$ emerged from the covering of α-Fe$_2$O$_3$ in SiO$_2$, indicating the bending and stretching modes of O–Si–O. The FTIR results confirmed the formation of coating on the α-Fe$_2$O$_3$ surface. Further calcinations could enhance the O-Si-O bond strength as well as improve the core and shell interactions. Figure 5c,d show the assessment results of the reddish pigments through the use of XPS. Figure 5c shows that Fe–O bonds and Si–O bonds are in the O1s pigment as evidenced from the high-resolution XPS spectrum. Meanwhile, a band that was observed at 103.5 eV in the Si 2p XPS spectrum is expected in pure silica.

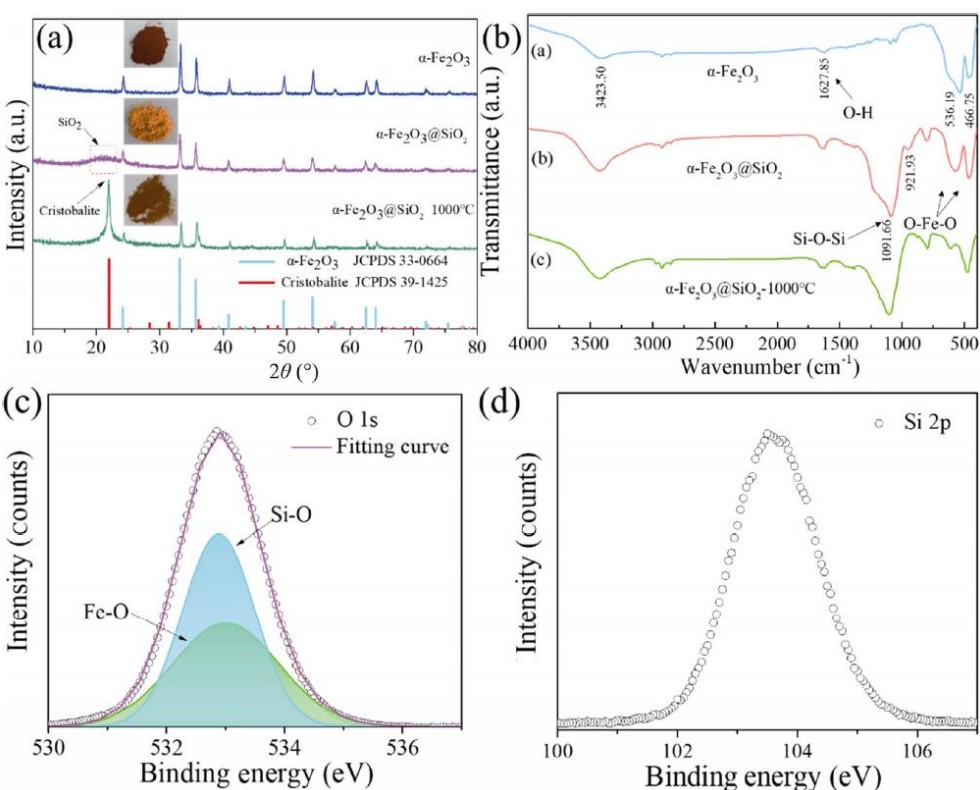

**Figure 5.** (**a**) XRD patterns and (**b**) FTIR spectra of different samples; high-resolution XPS spectra of (**c**) O 1s and (**d**) Si 2p for α-Fe$_2$O$_3$@SiO$_2$ pigments calcined at 1000 °C [88]. Reproduced with permission from Chen, S., et al., Applied Surface Science; published by Elsevier, 2020.

Li et al. [47] conducted an analysis on the synthesized γ-Ce$_2$S$_3$@SiO$_2$ core-shell materials using TEM test. The TEM images in Figure 6 show the silica shell being formed at various coating times. A clear layer covers the γ-Ce$_2$S$_3$ but it is not found on the samples that are not coated, which is in accordance to the SEM analysis. Figure 6b–d shows a correlation between the increasing thickness of the coating layer and increasing coating times. It was demonstrated that when the particles were coated once, twice and thrice times, the thickness was increased to 70 nm, 100 nm and 140 nm, respectively. This clearly indicated that it is possible to control the coating thickness through number of coatings being applied.

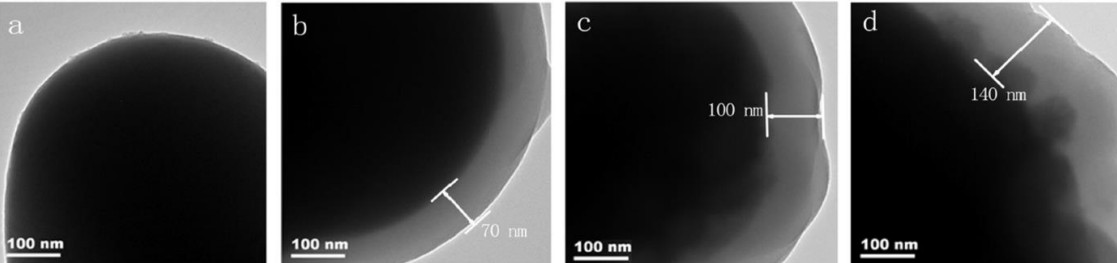

**Figure 6.** TEM images of (**a**) uncoated γ-Ce$_2$S$_3$ and (**b**) once, (**c**) twice (**d**) thrice coated γ-Ce$_2$S$_3$@SiO$_2$ core-shell particles [47]. Reproduced with permission from Li, Y.-M., et al., Surface and Coatings Technology; published by Elsevier, 2018.

Liu et al. [97] used four types of tests (FTIR, TEM, XRD and EDS) to assess the morphology of fabricated -Ce$_2$S$_3$@SiO$_2$ samples. The first step was to assess the SiO$_2$ thickness used to coat γ-Ce$_2$S$_3$. This was performed through the TEM test. Figure 7 shows different amounts of volume ratios of water/ethanol that was used for the preparation of uncoated γ-Ce$_2$S$_3$ pigments and SiO$_2$ xerogel coated γ-Ce$_2$S$_3$. Figure 7a presents the deposited surface with irregularly large chunks accompanied by small particles on the uncoated γ-Ce$_2$S$_3$ pigments. The detected Zn signals within EDS spectra indicated that the colour stability of the uncoated γ-Ce$_2$S$_3$ pigments can be controlled using ZnO. Another advantage of the application of these pigments is its low H$_2$S emissions. Figure 7b–d shows presence of core-shell structures within all the pigment particles during coating. Simultaneously, Si signal is detected as shown in Figure 7f. This proves that SiO$_2$ xerogel made up of the coating layer that is formed on the γ-Ce$_2$S$_3$ surface. Figure 7b,c on the other hand is showing that the application of water/ethanol volume ratio of 15/105 (48 nm) and 20/100 (60 nm) results in a moderately uniform shell size. However, Figure 7d shows that when the ratio is adjusted to 25/95, the thickness of the shell is no longer uniform. The main reason for this is that as water volume rise, it will accelerate TWOS hydrolysis. This means that during the coating process, shell thickness is no longer uniform because of the competition between surface and silica nuclei.

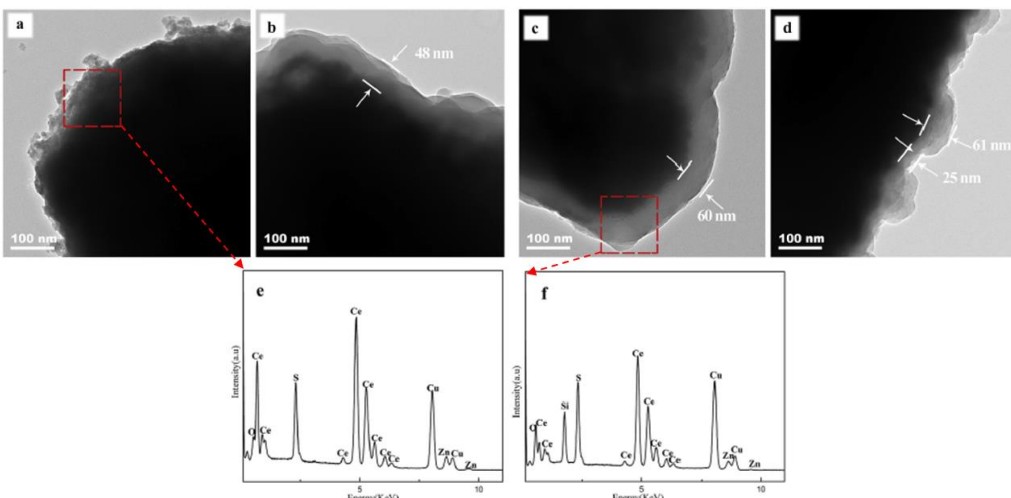

**Figure 7.** TEM images and EDS patterns of SiO$_2$ xerogel coated -Ce$_2$S$_3$ prepared with different water to ethanol ratio: (**a**) S0, (**b**) S1, (**c**) S2, (**d**) S3, (**e**) EDS spectra of S0 and (**f**) EDS spectra of S2 [97]. Reproduced with permission from Liu, S.-G. et al., Applied Surface Science; published by Elsevier, 2016.

Figure 8 shows the reflectance spectrum measured by Sadeghi-Niaraki et al. [98] for the as-produced Fe$_2$O$_3$@TiO$_2$ with crystallite size (nm) of CT (32.2 nm), CFT2 (31.4 nm),

CFT4 (28.4 nm), and CFT5 (13.3 nm) samples. CT sample showed that the reflectivity at wavelengths improved which was due to the increase in the crystallinity of the emergent rutile phase. After the calcinations, the reflectance value increased as the sample experiences the crystallisation. Figure 8c shows that the presence of $Fe_2O_3$ produced darker hues within the samples in addition to NIR reflectance being reduced. The NIR solar reflectance for the samples was CT (76%), CFT2 (73%), CFT4 (68.8%), CFT5 (68.4%) and CF (39.3%). Figure 8d shows the IR reflectance process within $Fe_2O_3$–$TiO_2$ and $Fe_2O_3$ particles.

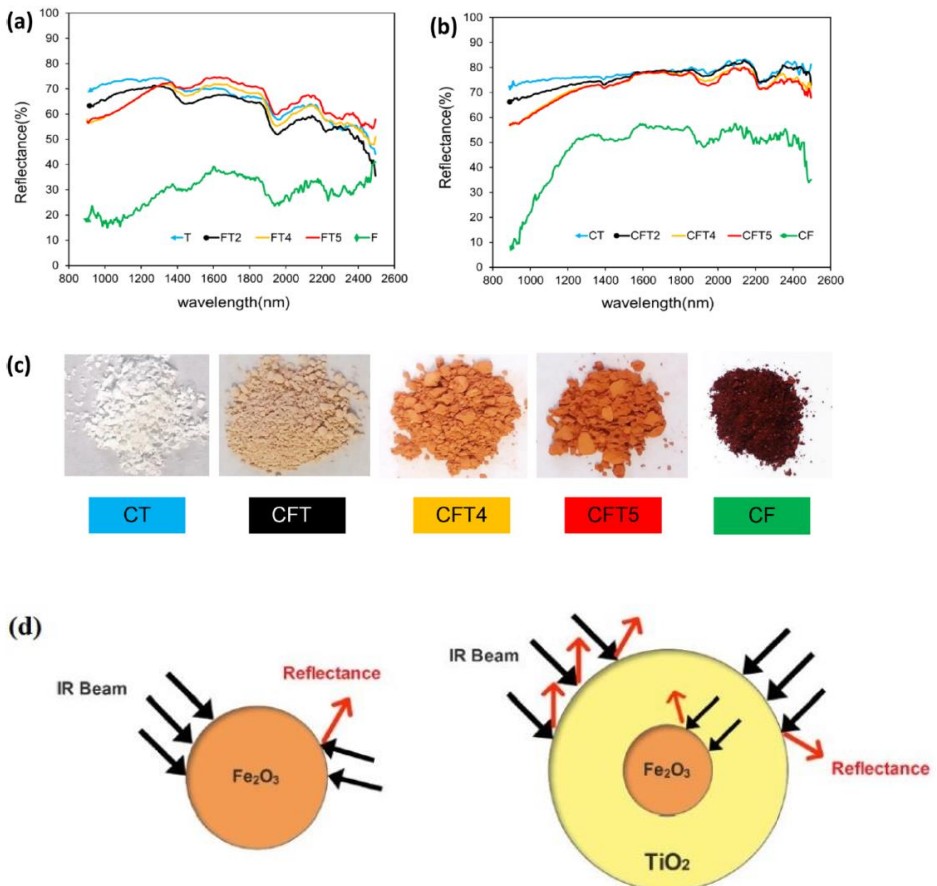

**Figure 8.** Reflectance spectra of (**a**) T, FT2, FT4, FT5 and F samples, (**b**) CT, CFT2, CFT4, CFT5 and CF samples, (**c**) photographs of CT, CFT2, CFT4, CFT5 and CF samples (**d**) proposed mechanism of IR reflectance in $Fe_2O_3$ and $Fe_2O_3$@$TiO_2$ composites [98]. Reproduced with permission from Sadeghi-Niaraki, S. et al., Materials Chemistry and Physics; published by Elsevier, 2019.

Li et al. [99] tested the high temperature tolerance of the red pigments made from $Ce_2S_3$@$SiO_2$-based core-shell NPs. Figure 9 shows the XRD patterns related to the γ-$Ce_2S_3$@c-$SiO_2$ samples, where their production was subjected to various calcination temperatures. There is an absence of the commonly found $SiO_2$ diffraction peaks when the calcination temperatures occur at the range from 1100 °C to 1150 °C. However, the diffraction peaks occurred during the γ-$Ce_2S_3$ crystalline phase. This means that $SiO_2$ failed to crystallize. However, c-$SiO_2$ diffraction peak initiated as the temperature reached 1200 °C. This suggests that $SiO_2$ will only crystallise within Ar gas atmosphere when temperature reaches 1200 °C. c-$SiO_2$ diffraction peak's intensity remains approximately at constant level as temperature is further raised to 1250 °C. Therefore, c-$SiO_2$ is prone to crystallisation when two conditions are met; (a) it is within Ar-gas atmosphere (b) temperature to be at least 1200 °C. In another study, Li et al. [100] analyzed the γ-$Ce_2S_3$ red pigments' resistance through XRD test.

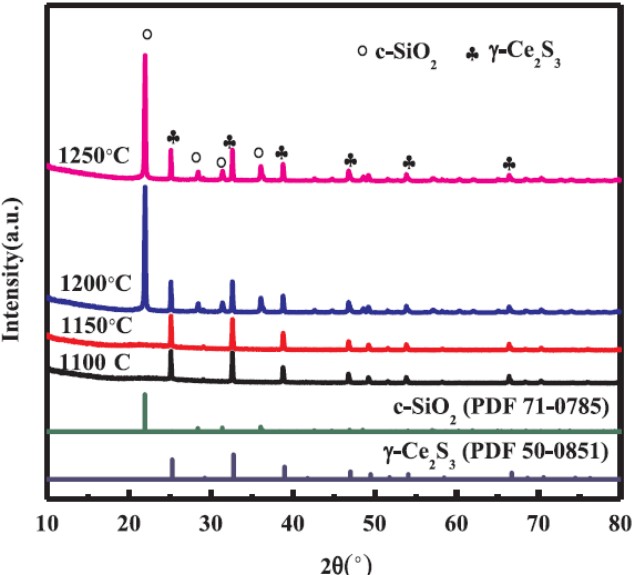

**Figure 9.** XRD patterns of the $\gamma$-Ce$_2$S$_3$@c-SiO$_2$ samples at different sintering temperatures in Ar gas atmosphere [99]. Reproduced with permission from Li, Y. et al., Applied Surface Science; published by Elsevier, 2020.

*3.3. Applications of Core-Shell Pigments*

Pigments can serve as decoration or delineation purposes in the public street thus improving both the aesthetics and public safety. Infrastructures that are well-built and well-planned can motivate individuals to use them such as walking, cycling or ease of access for Personal Mobility Devices (PMDs). If individuals are more willing to do the aforementioned activities, they are less likely to use their cars for short destination, which in return contribute to their overall health lifestyle. Figure 10 shows various red pigments being applied on Singapore roads for pedestrian use. Despite an advancement in the comprehension of the causes and effects of material failure, it remains a major concern in the entire construction industry. Exterior durability is typically enhanced through the use of high-performance coatings. Pigments are chosen for both the desired colour and performance [101]. The paint industry would exclusively use high-quality pigments. It is important for these pigments' particles to be homogenous in size as it could have an effect on the paint's attributes such as lightening capacity, hiding power, tinting strength and gloss. Furthermore, it is mandatory to apply nanoscale pigment particles in luminescent materials for the purpose of UV-coatings and colouring.

There is a higher desirability for coloured asphalt and red concrete in comparison to traditional materials as the former has better aesthetics from the viewpoint in architecture design [102]. The past few years have seen major development in nanomaterials and nanotechnology. This has made synthesising core-shell NPs possible, which also contributed to developing pigments that are sustainable yet higher colour stability as well as able to tolerate harshness. The development of the pigments with increased durability has led to increase many potential applications in applying colours onto concrete and asphalt. This leads to further development within the architect industry where they have the options to apply colours that carry more stability and higher tolerance to abrasion. Such developments indeed could be combined with the aesthetic and decorative aspects of conventional concrete thus forming an additional material with attractive features.

## 4. Nano-Enhanced Phase Change Materials

Buildings consume about 45% of global energy. Many passive cooling methods have been used to lower the consumption rate. In addition, the phase change materials (PCM) are installed within these buildings for the purpose of promoting temperature moderation, stopping heat from accumulation, improved heat absorption and minimize indoor heat

gain. The method in which PCM stores thermal energy is effective in improving the buildings' aggregate heat capacity. Interest has been strong in PCMs that has high energy density to be deployed in buildings with high thermal inertia in order to save a high amount of energy. PCMs have their own drawbacks and the primary one being extra time required to charge/discharge energy process as well as storage performance, which happens due to poor thermal conductivity. Therefore, attention is focused on improving their thermal conductivity through the use of nanotechnology and nanomaterials. There has been a rapid development lately within the nanomaterials field resulting in the latest technology with nanosized particles in improving the PCM's thermophysical properties. PCM has several thermal and physical qualities such as viscosity, heat capacity, super-cooling and thermal conductivities. These attributes could be significantly improved through dispersal of thermal conductive nanoparticles including nanometal-oxide, nanocarbon and nanometals. This article explores the research that have been recently conducted in the aforementioned development of nanomaterials that are being used to improve the PCMs thermal performance. This is appropriate in passive-cooling within the built-environment. The focus would be on materials' type, method of synthetisation, and the outcome of the improvement.

According to Mardiana and Riffat [103], about 30% of the total energy of any nation is consumed by the residential, institutional, commercial and industrial buildings. Approximately, 60% of the energy is used in a building equipped with heating, ventilation and air-conditioning (HVAC) systems. PCM is a preferred building cooling method in comparison with other methods as it compliments green building with efficient energy performance [104]. An effective strategy is phase change technology, where it could enhance the building's thermal mass. This means removing heat from indoors, reduce temperature variations and disperse heat away from the building with the overall impact of increasing the comfort of the occupants. Studies have discovered that PCMs energy saving ranged from 10% to 30% from air-conditioning consumption within various climate in the United States [105]. During the summer, the energy savings could be up to 30% when PCMs are built on building walls. Microcapsules of PCM application results in the reduction of internal temperature of a building by 4 °C and in a longer period of time, it stops the temperature from reaching for more than 28 °C.

PCMs are classified as inorganic, organic and eutectic. Types of inorganic PCMs include metal alloys, metals, and hydrated salts whereas an example of organic PCMs is hydrocarbons-based paraffin wax. There are disadvantages of PCM such as thermal instability, corrosive property, sub-cooling, low thermal conductivity, leakage, phase segregation and many more [106]. In comparison, organic PCMs are sometimes more suitable due to their non-corrosive properties, immense latent heat capacity, congruent melting and self-nucleation, chemically inert as well as being thermally stable [107]. Dispersion of a controlled amount of nucleating or dispersant agents is a solution in addressing subcooling and phase segregation issue [106]. Nevertheless, PCM has an inherent low thermal conductivity, denoted by "k". These results in low level of responsiveness during which a thermal change occurs rapidly due to charging/discharging process and its lowered storage capacity. Such issue becomes the centre of attention in research related to thermal energy storage. The k values of hydrocarbon-based PCM range from 0.1 to 0.4 W/mK. Noctadecane is a type of PCM, which possess low solid state thermal conductivity at 0.35 W/mK. It's liquid state however is at 0.149 W/mK [108].

Rapid development of nanomaterials led to the emergence of novel application strategy at its high level of conductive ultra-small nanosized particles including metal oxides, carbon and metals. These can be used to produce nano-enhanced PCM (nePCM) with significant micro-convection [3] and thermal conductivity [109]. Ample opportunities exist for nanomaterials potential applications in the cutting edge phase change technology. PCM has generated intense interest in the application of nanometer-scaled thermal conductors through nanofibers, nanoparticles, nanosheets, nanotubes and nanofoams [104]. The thermal conductivity of PCM can be enhanced using three methods. First method involves

the incorporation of PCM into porous media such as metallic foams and porous carbon, which has high thermal conductivity. Second method deals with the dispersion of high thermal conductivity metallic nanostructures or nanoparticles of Cu, Ag or Al to the PCM. Third method deals with the microencapsulation of the PCM [108]. The thermal conductivity and strength of microcapsules' wall could be increased through nanoparticles that are made of silver [110]. An efficient way of improving PCM additive is copper particles due to its high conductivity and low cost [111].

Three types of elements with thermal conductivity have extensively been studied [112]. These include carbon-based nanostructures such as graphene nanoflakes, nanoplatelets, carbon nanotubes CNT and nanofibers; metallic oxide like $TiO_2$ and MgO; metals like Al, Ag and Cu. There is a significant improvement on heat transfer through the use of nanoparticles. The nanoparticles that can be applied to achieve this are carbon that possess various morphologies such as ceramic oxide ($CuO$, $Al_2O_3$), metallic nitrides (AIN, SiN), metallic carbides (SiC) and stable metals (gold Au) [105]. Nanomaterials that comprise of metals (Cu, Ag and Al), metal oxides (ZnO) and carbon (single wall SWCNT, graphene nanosheets, active carbon, carbon nanofibers, expanded graphite sheets) increase PCM's rate of heat transfer [113]. In this view, the prominent research being conducted on the development of thermal conductivity through the dispersion of three primary PCM nano-enhancers such as nanometals, nanocarbons and nano-metal oxides.

### 4.1. Nano-Metal Enhancer

It is a common knowledge that metal is efficient at heat conductivity. Silver in particular is the optimal conductor of heat and electricity in comparison with other metals. Its thermal conduction value is approximately 430 W/(mK). The next two metals that are close to silver in terms of thermal conductivities are copper and gold. Gold and silver have two major disadvantages, vulnerable to oxidation and high cost. Therefore, copper, at a significantly lower cost has the advantages in comparison. Despite this, all the three aforementioned metals have been extensively researched as possible solutions in addressing PCM's thermal conductivities. Al-Shannaq [108] improved the PCM's thermal conductivity (k) by 1168% through the use of nano-thick Ag shells. Specially microencapsulated pure PCM could be used to address leakages issues during its change of state from solid to liquid. However, the microencapsulated shell with poor conductivity value k served as a barrier to achieving a desirable level of heat transfer and energy storage. A method has been formulated to enhance the PCM's microencapsulated k value that involved the use of a layer of metallic shell to cover the microcapsules. This was done by activating the surface with dopamine and conducting electroless plating. The k value was increased to 0.189 from 0.062 W/mK when the diameter of uncoated PCM was increased to 26.9 μm from 2.4 μm. While the diameter was retained at 26.9 μm, a significant increase of the thermal conductivity (about 1168%) of metal-coated PCM capsules (2.41 W/mk from 0.189) was achieved. Such improvement of the thermal conductivity is highly correlated with the size of the shell area that is coated with silver on the surface of the PCM microsphere. The rapid improvement occurs upon the formation of the thermal conduction pathways.

Deng et al. [114] have made another significant improvement (1030%) in the thermal conductivity of the PCM via the synthesis of AgNWs. First, shape stabilised phase change materials (polyethylene glycol-silver/EVM ss-CPCMs) composites were produced via the embedment of PEG-Ag nanowires into expanded vermiculite EVM. To prevent the PCM leakage as well as to improve its thermal conductivity, a technique was proposed whereby the mixing and embedding are performed mechanically. For the purpose of PCM latent energy storage, polyethylene glycol was used. Figure 10 shows the silver nanowires that served as the thermal conductivity promoter. Furthermore, the PCM leakage during the melting was addressed through a support matrix (EVM vermiculite), enabling the enhancement in the mechanical strength.

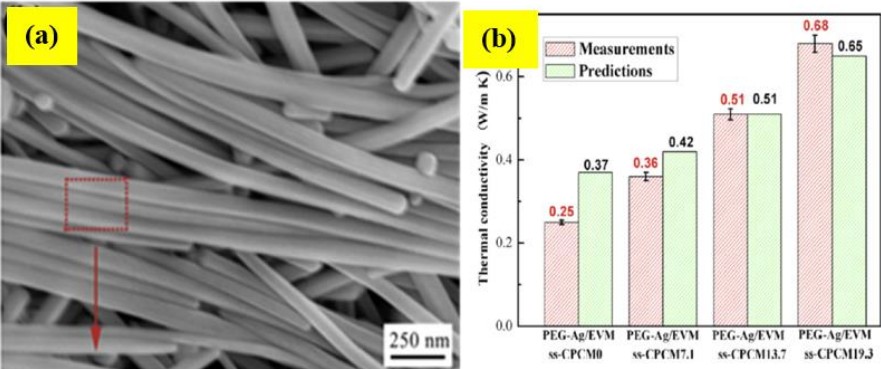

**Figure 10.** (**a**) SEM photos of synthesized silver nanowires. (**b**) Variation between the predicted thermal conductivity k value with measured values of PCM nanocomposites [114]. Reproduced with permission from Deng, Y. et al., Chemical Engineering Journal; published by Elsevier, 2016.

Significant improvement in the k value of PEG infused silver vermiculite composites was achieved using nanowires of length 5–20 μm and diameter 50–100 nm. An increase as much as 1130% for the k value (0.68 W/mK) was achieved compared to the neat PCM with latent heat capacity at 96.4 J/g. The vermiculite has incited supercooling to occur where the temperature dropped by 7 °C upon the PCM for PEG–Ag/EVM ss-CPCMs. Such reaction is similar to nonuniform impregnates for developing nucleation and promoting the formation of PEG-crystal. Such improvements are as a result of high k values due to the dispersion of silver nanowire and vermiculite. Zeng et al. [107] obtained about 800% improvement in the thermal conductivity using CuNWs. The premise of the research is to explore the impact of CuNWs, which is copper nanowires has upon the tetradecanoyl (TD)'s k value as the phase change material. The TD was synthesised and classified accordingly to the range of weight fractions of CuNW. The ratio and diameter of free-standing copper nanowires were at 350–450 and 90–120 nm, respectively with 40–50 μm in length. The CuNW can then be fabricated in bulk through simple technique involving chemical reduction that is water based at room temperature.

Figure 11 presents the SEM images of the composite results, demonstrating that CuNws in TD has decent dispersion and entanglement. It is worth noting that the rate of weight loss is lower in comparison to pristine TD due to the structural nature of CuNWs, which is similar to a sponge and is capable of storing the TD within the voids. When the CuNWs is increased by 58.9 wt%, the thermal conductivity increased up to nine-fold, an 800% enhancement.

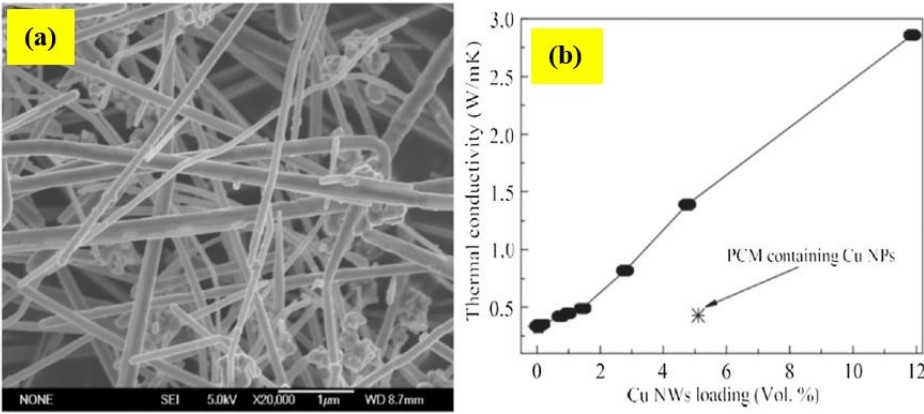

**Figure 11.** (**a**) SEM photos of synthesized CuNWs. (**b**) Thermal conductivity of PCM composites with increasing CuNWs loadings [107]. Reproduced with permission from Zeng, J.-L. et al., Solar Energy Materials and Solar Cells; published by Elsevier, 2012.

Zeng et al. [115] improved the thermal conductivity by about 356% using AgNWs (380%). The experiment involved the synthesis of silver nanowires and production of silver-doped PCM nanocomposites. The inclusion of AgNWs at 45 wt% results in two to three times enhancement of thermal conductivity in graphene-doped PCM. The enthalpy is reduced by 50% and its heat storage capacity has also been reduced. In terms of size, graphene dopants are ten times smaller in comparison to doping with silver nanowires. Furthermore, the enthalpy value has also been reduced three times in comparison to AgNWs. Shah et al. [116] have increased the PCM thermal conductivity by 160% through the use of copper nanowires (CuNWs). The enhancement of thermal conductivity (more than 50%) of calcium chloride hexahydrate is achieved by adding a trace of CuNWs at 0.17 wt%. The use of nano-copper results in optimum enhancement of k value at 160%; or an increase to 0.564 W/mK of PCM composite in comparison to 0.217 W/mk of neat PCM. Just a trace of CuNWs could result in such a significant improvement thus nanoadditives can be considered as cost efficient when being applied in buildings.

Molefiet et al. [117] showed 70% improvement in the thermal conductivity using CuNPs. The thermal conductivity of paraffin was increased almost linearly as the CuNPs amounts were increased. Paraffin wax was used as the PCM base, which was subsequently mixed with molecular-weight polyethylene at low, medium and high rate. The copper particles were mixed with paraffin mixture resulting in the enhancement of the base polyethylene PCM's k value. Tang et al. [118] improved the thermal conductivity by 38.1% using CuNPs based on $SiO_2$ -embedded-PEG PCM composite that is shape stable. When 2.1 wt% CuNPs were added, the k value was increased by 38.1% in comparison to neat PCM. Further addition of copper nanoadditives results in improvement on PEG/$SiO_2$ hybrid PCMs. Wu et al. [119] have made 30.3% improvement on thermal conductivity through the use of CuNPs. Their results have shown a correlation where 1wt% of CuNPs could decrease the paraffin PCM heating by 30.3% and cooling by 28.2%. The charging time decreased by 30.3% while the discharging time was decreased by 28.2% upon the doping of nanocopper particles into the nanocomposites with 1 wt%. Melting PCM heat transfer rate is enhanced through the addition and mixture of nanoadditives (aluminium, copper and copper/carbon nanomaterials). In terms of improvement on heat transfer, nanocopper particles offer the most significant rate amongst others.

*4.2. Nano-Metal Oxide Enhancer*

Two examples of good heat conductors are alumina and copper, both of which are metal oxides with values from 30 to 40 W/mK. Pure metals typically are better heat conductors but they are not as chemically stable in comparison to the metal oxides. In addition, metal oxides are more cost effective and reliable in its performance. For these reasons, they are more sought after as a material to replace pure metals. Babapoor et al. [120] used various NPs types to enhance the thermal conductivity of k value. The metals with the enhancement percentage of $Al_2O_3$ (144%), $Fe_2O_3$ (144%), ZnO (110%) and $SiO_2$ (110%) were obtained. In these tests, nanomaterials of silica (~20 nm), alumina (~20 nm), iron oxide (~20 nm), and zinc oxide (>50 nm) were used. These nanomaterials were added as thermal enhancers and mixed with NPs (SDS) as well as surfactant (CTAB) to achieve enhanced PCM. The sample doped with $Al_2O_3$ NPs showed the highest enhancement in the thermal conductivity of 0.919 W/mK.

The doping of NPs gave various enhancement (%) level depending on the concentration (wt%) of $Al_2O_3$ NPs: 4 wt% (120%), 6 wt% (141.2%) and 8 wt% (144%); $Fe_2O_3$ NPs of 4 wt% (80%), 6 wt% (135%) and 8 wt% (144%); ZnO NPs of 4 wt% (85%), 6 wt% (100%) and 8 wt% (110%); $SiO_2$ NPs of 4 wt% (78%), 6 wt% (110%) and 8 wt% (110%). The results revealed that higher level of concentration of conductive nanomaterials lead to higher k value of the nanocomposites. It was concluded that $Al_2O_3$ and $Fe_2O_3$ carry the most significant impact in terms of enhancing the thermal conductivity of paraffin-based PCM. Sharma et al. [121] achieved an improvement in the thermal conductivity of about 80% using $TiO_2$. The study involved the performance of palmitic acid (PA) based thermal

energy storage of synthesised PCM composites that were doped with $TiO_2$ NPs. By mixing $TiO_2$ into neat PCM an enhancement in the k value was 12.7%, 20.6%, 46.6% and 80% for the corresponding $TiO_2$ concentrations of 0.5 wt%, 1 wt%, 3 wt% and 5 wt%, respectively. The high concentration of $TiO_2$ within the PCM resulted in curvilinear characteristic of thermal enhancement. Li et al. [122] achieved 43.8% and 404% improvement in the thermal conductivity using $TiO_2$ NPs foam and $TiO_2$ NPs with a nanocarbon shell layer, respectively. The synthesis of porous $TiO_2$ foams PTFs involved the use of octane as microemulsifier and $TiO_2$ as particle stabilizer (microemulsion technique) as shown in Figure 12. The nanosized $TiO_2$ measured at approximately 23 nm consisting of 20% rutile and 80% anatase. Polyacrylic acid-ammonium salt is used as the dispersing agent. It is added on the surface modifier along with a small amphiphilic molecule propyl gallate ($C_{10}H_{12}O_5$).

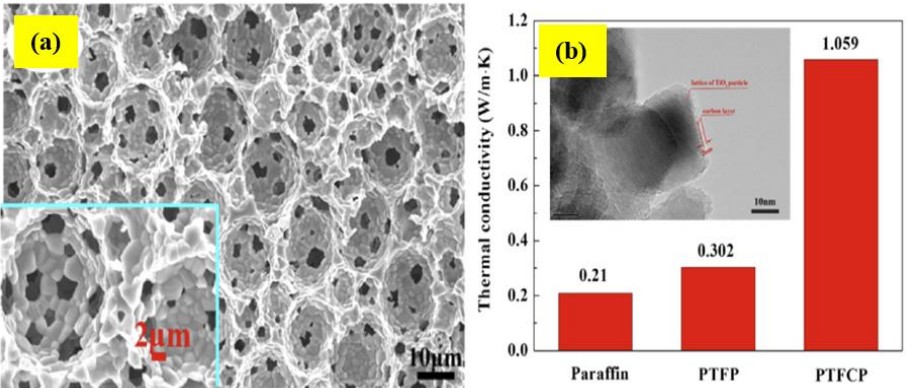

**Figure 12.** (**a**) SEM images of PTF. (**b**) Enhancement of thermal conductivities of PCM paraffin, PTF/PCM, and PTFC/PCM composites by 43.8 and 404%. (Inset) TEM photo of the prepared PCM composite carbonized porous $TiO_2$ foams (PTFC) particles [122]. Reproduced with permission from Li, Y. et al., Applied Energy; published by Elsevier, 2016.

The 3D porous structure of PTFs contains continuously connected holes, enabling the full absorption of paraffin wax without the need of any surfactant. The structure can also absorb sucrose and can burn off at 1200 °C, resulting in a thin carbon-based film wherein the carbon nanolayer is only 2 nm thick. Both pure PTF and carbon-based PTF nanocomposites were more conductive than pure paraffin with k values of 0.302 and 1.059 W/mK, respectively. The k value of pure PCM reached to 0.302 W/mK when 25 wt% of $TiO_2$ was added. This clearly indicated that the addition of $TiO_2$ can enhance the k value by 0.092 W/mK. $TiO_2$ foam structure lined with carbon nanofilm plus paraffin showed a k value of 1.059 W/m K, indicating an increase of 504% than pure paraffin. This significant increase was mainly due to the carbon matrix adherence onto the $TiO_2$ NPs surfaces. It was affirmed that the novel hybrid of $TiO_2$ NPs-porous foam with inner-lining carbon nanofilms is effective for the enhancement of PCM demanded in the industrial purposes. Zhang et al. [123] made about 18.2% improvement in the thermal conductivity using $TiO_2$ wherein a novel thermal-insulating film and polyvinyl-chloride (PVC) film matrix were incorporated. Both $TiO_2$ and microencapsulated n-octadecane PCM were used to block UV and act as an additive to regulate the temperature. When $TiO_2$ NPs were added at 6 wt%, the k value of the pure micro-PCM was reached to 0.2356 W/mK from 0.1994 W/mK for the matrix, indicating an increase by 18.2%. Such thin film with excellent heat insulation and thermal regulating properties were affirmed to be useful for the indoor living spaces and cars.

Sahan et al. [124] achieved about 60% of thermal conductivity improvement using sol-gel synthesized $Fe_3O_4$ NPs. These $Fe_3O_4$ NPs (diameters ranged from 40 to 70 nm) were prepared using iron chloride hydrates (($FeCl_3$ $6H_2O$, $FeCl_2$ $4H_2O$), hydrochloride and ammonia. They were mixed with paraffin in two concentration levels (10 and 20 wt%). Particle aggregation was minimised through surface capping of oleic acid. These $Fe_3O_4$

NPs were uniformly dispersed on the paraffin matrix. The results showed an improvement in the k values by 48% and 60% for the corresponding NPs concentration of 10 wt% and 20 wt%, respectively. This showed that nanomagnetite particles doping in PCM was very effective towards the improvement of its thermal conductivity and cost. Jiang et al. [125] observed 55% improvement in the thermal conductivity of PCM using nano-$Al_2O_3$. The microencapsulation of paraffin was responsible for the formation of poly(methylmethacrylate-co-methylacrylate) polymeric PCM microcapsules (MEPCM). These microcapsules were further added with alumina NPs via the emulsion polymerization, causing significant enhancement of k value (Figure 13) from 0.245 W/mK to 0.38 W/mK (increase by 55%). There is near parity in terms of the enhancement rate and dosage of nano-$Al_2O_3$, indicating that the presence of nano-$Al_2O_3$ caused a higher thermal conductivity increase of PCM microcapsules.

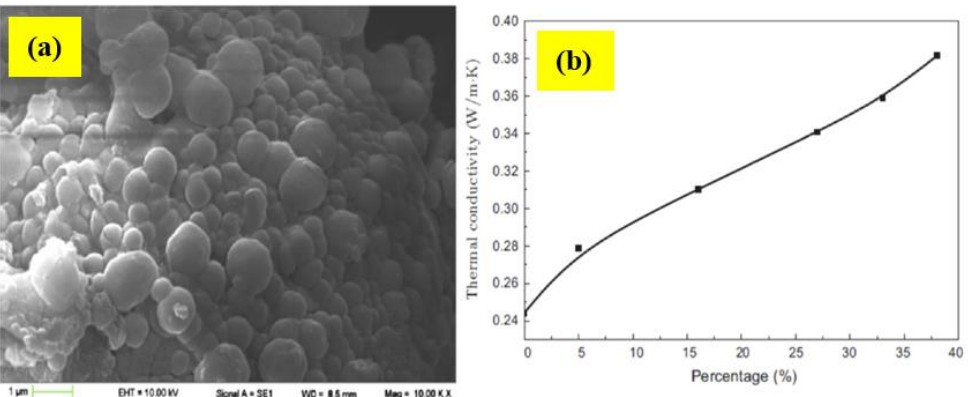

**Figure 13.** (**a**) SEM photos of PCM microcapsules with 27 wt% nano-alumina. (**b**) Thermal conductivities of PCM with various contents of nano-alumina [125]. Reproduced with permission from Jiang, X. et al., Applied Energy; published by Elsevier, 2015.

Tong et al. [126] improved the PCM's thermal conductivity using nano-$SiO_2$. The polymeric melamine-urea-formaldehyde was used for the polymerisation of in situ PCM paraffin microcapsules before adding graphite and nano-$SiO_2$. The results revealed that the successful rate of paraffin microencapsulation was at 80% wherein the PCM paraffin was able to sustain its thermophysical properties. The addition of nano-$SiO_2$ could change the microcapsules resistance against high temperature, reinforcing the structural strength of composite and high affinity to water. The k value was improved significantly during melting time after the nanomaterials were added. Ai et al. [127] enhanced the thermal conductivity of PCM using high energy planetary milling wherein $ZrO_2$ nanopowder-based stearic acid PCM was developed. A new parameter called heat capability factor (HCF) was explored. Chloroform was used to disperse the nano-$ZrO_2$ PCM composites, providing a better alternative (than carbon tetrachloride) for the dispersion during $ZrO_2$ synthesis. The results revealed that chloroform could improve the surface morphology and spherodization of $ZrO_2$. The highest HCF value of 0.9 for the mean size of PCM particles was 1.2 μm. However, the HCF value was reduced to 0.3 when the mean size of PCM particles became 0.4 μm. The optimum PCM particles' size (1.2 μm) gave a significant enhancement in the heat storage capability of chloroform-treated composite $ZrO_2$-PCM particles. Song et al. [128] used MgOH NPs and made nePCMs to enhancing the fire resistance of PCM. The supporting materials used were nano-sized red phosphorus (RP), MgOH and ethylene propylenedieneter polymer plastic (EPDM). The observed increase in the fire resistance quality was ascribed to the magnesium hydroxide within the flame retardant shape-stable PCM composite. It was argued that the fire-resistant attributes of the PCM can be further improved through the reduction of NPs diameter. Consequently, larger surface to volume ratio of MgOH can produce rapid breakdown and high reactivity when

subjected to the combustion process, indicating higher fire resistance quality attainment of PCM composite.

### 4.3. Nano-Carbon Enhancer

Carbon has higher thermal conductivity when benchmarked against metals and metal oxides. Graphite, graphene and CNTs thermal conductivities can be up to five times higher than silver. Research studies have increasingly focused on the carbon nanomaterials thermal conductivities due to their continuous decrease in production cost. Ji et al. [129] improved PCM's thermal conductivity by 1700% using ultra-thin graphite foams (UGF). The k value was increased by 18 times after adding UGF (at approximately 1.2 vol%) into the PCM matrix. However, no changes in the specific heat fusion or melting temperature were observed. Graphite foams consisted of ultrathin graphite connected strips. These strips possessed a higher k value than metals and solid carbon foams, indicating their better heat response and thermal properties. Liang et al. [130] obtained 1300% improvement in the thermal conductivity using superoleophilic graphene nanosheets mixed with porous nickel Ni foam. In the synthesis of polydimethylsiloxane (PDMS-G-NF) modified graphene-covered nickel foam they used graphene nanosheets layering onto the porous Ni foam surface, causing the formation of graphene-nickel foam G-NF. Further modifications were performed on the G-NF surface support using siloxane PDMS for the fabrication of shape-stable PCM composite.

Chen et al. [131] achieved 500% improvement in the k value of PCM using CNT foam. The PCM was absorbed by a permeable support matrix, a carbon nanotube network with structure similar to sponge. The heat storage capacity of PCM was improved and became efficient for both heat and electricity conduction. In addition, the PCM composite could absorb light energy and generate heat via electricity. Figure 14 shows the PCM composite consisted of paraffin filled soft-flexible CNT-based porous material. The support matrix that is deformable has high rate of thermal conductivity during the solidification and melting processes.

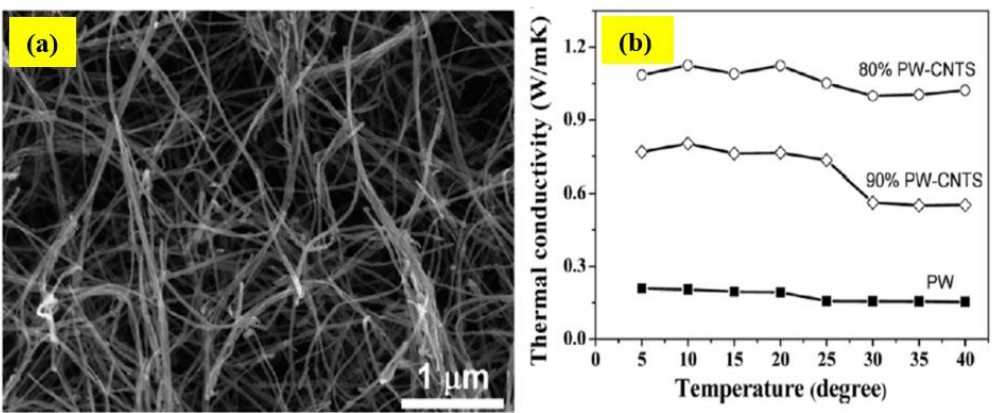

**Figure 14.** (**a**) SEM photo of the interior of CNT porous foam revealing a highly sponge-like microstructure. (**b**) Thermal conductivities of neat paraffin wax with 10 and 20 wt% loadings of CNT foams, i.e., 80 and 90 wt% paraffin [131]. Reproduced with permission from Chen, L. et al., ACS Nano; published by American Chemical Society, 2012.

Shi et al. [132] used exfoliated graphite nanoplatelets (xGnP) and grapheme and improved the corresponding k values by 1000% and 100%, respectively. Such improvement resulted in the formation of paraffin PCM materials that are stable. Approximately 2 wt% of the graphene was added to paraffin and heated to around 185 °C. The paraffin retained its form despite reaching significantly high melting point. It was claimed that to decrease the cost, trace amount of graphene and xGnP can be doped together thereby improving both stability and heat dissipation of PCMs. xGnP-doped PCM resulted in a k value of 2.7 W/mK which was considerably higher than graphene-doped PCM (approximately

0.5 W/mK) and neat paraffin (0.25 W/mK). Wang J. [133] achieved 305% improvement in the PCM thermal conductivity by adding carbon nanofibers (CNFs) as nanofillers into the palmitic acid (PA). The phase temperature change was approximately 62.5 °C after the addition of unwashed acid. The range of length and diameter of the CNFs was 200–500 nm and 5–50 µm, respectively (Figure 15). Alkali potassium hydroxide (KOH) was used to chemically treat CNFs, reducing the thermal boundary resistance of the fibre matrix.

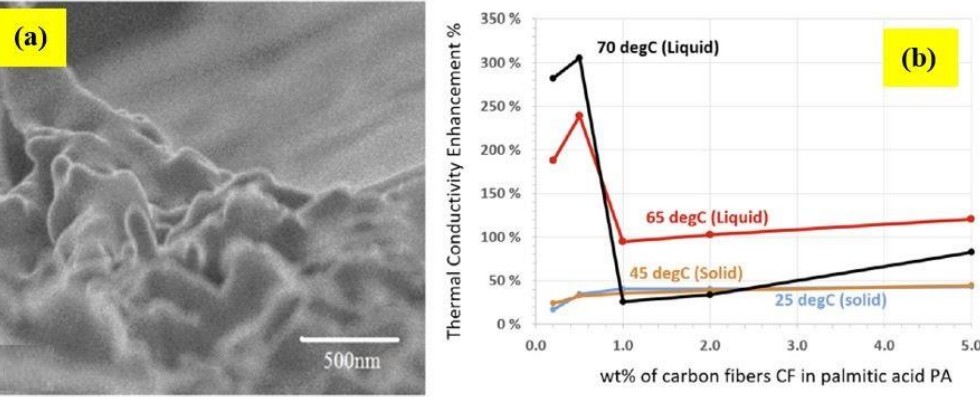

**Figure 15.** (**a**) SEM image of M-CNF/PA with 1.0 wt% M-CNF. (**b**) Thermal conductivity enhancement with 0.2, 0.5, 1, 2 and 5 wt% of carbon nanofibers (CNF) in palmatic acid (PA) [133]. Reproduced with permission from Wang, J. et al., Journal of Applied Physics; published by AIP Publishing, 2011.

Cui et al. [134] improved the PCM's thermal conductivity by 44% and 24% using nanofillers of CNF and MWCNT, respectively. The synthesis of the composite involved carbon fibres or nanotubes dispersion within both soy wax and paraffin (1, 2, 5 and 10 wt%) at 60 °C. This proved that the nanofibers as additive can increase the paraffin k values significantly. The k value of pure paraffin and PCM composite (at CNF loadings of 10 wt%) were 0.320 W/mK and 0.450 W/mK, respectively. Wang et al. [135] improved the thermal conductivity of PCM by 46% using multiwalled carbon nanotubes (MWCNT). The traditional ball milling method was used to synthesize the MWCNT-PCM composites added with KOH. This method could improve its dispersion in palmitic acid. The stability and homogeneity of PCM composites were improved by modifying the grafted OH groups into the MWCNT surfaces. The MWCNT-palmitic acid composites with 1 wt% of MWCNT loading was shown to increase the k values by 46.0% and 38.0% on solid state at 25 °C and liquid state at 65 °C, respectively.

## 5. Nanopolymer Advanced Composites

THE definition of polymer nanocomposites (PNCs) is the combination of more than one material. In addition, the matrix consists of a polymer with the dispersed phase that possess a minimum of one dimension smaller than 100 nm [136]. Many decades of observation have deduced that incorporating nanofillers in small quantities within the polymer resulted in many improvements on its characteristics such as thermal, barrier, mechanical and flame-retardant properties while its processing is unaffected [137]. The optimum nanocomposite design necessitates the individual nanoparticles to disperse homogeneously within a matrix polymer. The main challenge in terms of dispersion state of nanoparticles is to achieve all the possible enhancements of its properties [137]. There is a potential for the nanofillers' uniform dispersion to result in significant interfacial area between the nanocomposites' constituents [137]. There are various factors that influence the reinforcing effect mainly polymer matrix properties, type and nature of nanofiller as well as polymer and filler concentration. Other factors focusing on the particle includes its size, aspect ratio, orientation and distribution [138]. There have been numerous types of nanoparticles being used to form the nanocomposites with various polymers including clays [138], carbon nanotubes [139], graphene [140], nanocellulose [141] and halloysite [142].

It is essential to evaluate the nanofiller dispersion within the polymer matrix. This is because there is a strong correlation between both the mechanical and thermal properties with the outcome of morphologies. The degree nanoparticles separation would result in three possible morphologies outcome [143] namely intercalated nanocomposites, conventional composites (also known as microcomposites) and exfoliated nanocomposites (Figure 16). In an event where the polymer is not intercalating between the layers of the silicate, the outcome would be separate phases of composite where its properties are within the same range as seen in conventional composites [144].

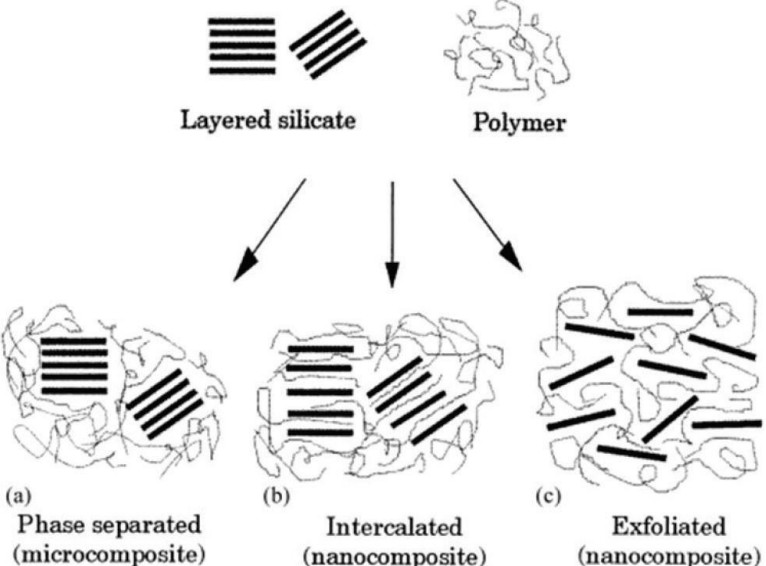

**Figure 16.** Possible structures of polymer nanocomposites using layered nanoclays: (**a**) microcomposite, (**b**) intercalated nanocomposite and (**c**) exfoliated nanocomposite [143]. Reproduced with permission from Alexandre, M. et al., Materials Science and Engineering: R: Reports; published by Elsevier, 2000.

An intercalated structure encompasses with at least one extended polymer chain, where it intercalates between the silicate layers. The outcome is a consistent order of multilayer morphology with polymer and clay layers that are intercalated. Exfoliated structure would result in the event of complete and orderly dispersion of silicate layers within a continuous polymer matrix [143]. Exfoliated nanocomposites have a large surface contact area between the nanoparticles and matrix. Such is one of the significant differences between conventional composites and nanocomposites.

*5.1. Compatibilization in Polymer Nanocomposites*

Compatibilization is of paramount importance to achieve a mixture of polymer or nanocomposite with the desired properties. Therefore, poor properties are attributable to the chemical nature differences between the polymers or polymer matrix with the NPs [145]. As previously mentioned, compatibilization is a significant factor in obtaining the desired properties. Degradation should be kept at a low probability and it occurs when the organomodifier is decomposed and when degradation products and polymers are interacting with each other. All of these have a significant influence upon the properties and morphology of the material [146] (Figure 17). There are three methods of productions for polymer nanocomposites; in situ polymerization, solution and melt blending. The production method is chosen based on the polymeric matrix type, nanofiller and the final products' desired properties [15].

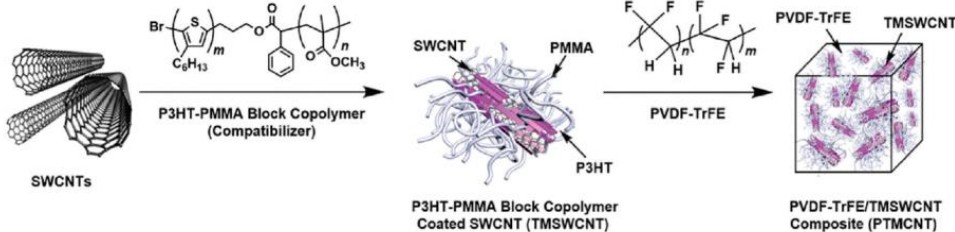

**Figure 17.** Schematic of compatible PVDF/SWCNT nanocomposite production [146]. Reproduced with permission from Cho, K.Y. et al., Composites Science and Technology; published by Elsevier, 2018.

*5.2. In-Situ Polymerization*

In-situ polymerization involves a correct dispersion of the nanofiller within the monomer solution prior to the beginning of polymerization process. This is to ascertain the formation of the polymer between the NPs. There are various methods of initiating polymerization such as heat, utilising the correct initiator, etc [147]. Such method could be used to achieve a polymer grafted NPs and high loading nanofillers with the absent of aggregation [148]. It is possible to include organic modifiers in order to assist the NPs dispersion and to be included within the polymerization [149]. Such method could be deemed as an alternative in producing nanocomposites through the use of polymers that may be deemed unstable thermally or non-soluble [150]. There are occasions where such method is applicable in solvent-free form [151]. Furthermore, such method may increase the performance of the products [152]. Mini-emulsion polymerization is dependent on the monomer droplets being produced, which are subsequently dispersed into a solution within a nanoscale [153]. The advantages include particle morphology that can be controlled [154], high functioning interfacial adhesion of the nanofillers [155] and higher transparency value [156]. This method could potentially [157] use higher nanofillers with no presence of agglomeration, increased performance of the final products, products with solvent-free form, outcome of covalent bond within the NPs functional groups and polymer chains as well as utilising the thermoplastic and thermoset polymers. A major disadvantage of such method is the agglomeration easing [148,150].

*5.3. Solution Blending*

Blending is the most used method because it is simple in terms of producing polymer nanocomposites. In comparison with other methods however, this method has higher difficulty in terms of achieving proper nanofiller dispersion within the polymer matrix [157]. Solution blending is a system that encompasses both the polymer and nanofiller that can be dispersed within a suitable solvent without much difficulty [147]. The dispersion of the nanofiller within the polymer can be achieved through magnetic stirring, ultrasonic irradiation or shear mixing [148]. Figure 18 demonstrates the use of this method, wherein the NPs are still dispersed within the polymer chains after the solvent evaporates. This nanocomposite that has just been produced could be developed into a thin film [157].

The solution blending posed a few constraints in economic and environmental terms. Thus, there is a need for an optimum method to achieve the desired product while addressing the constraints accordingly [158]. The advantages of solution blending include reduced gases permeability [159], simple operation and the use of conventional method for nanofillers of all types as well as the thermoset polymers and thermoplastic polymers [160]. The disadvantages include environmental and aggregation issues [158]. However, this method is restricted to water soluble polymers [161].

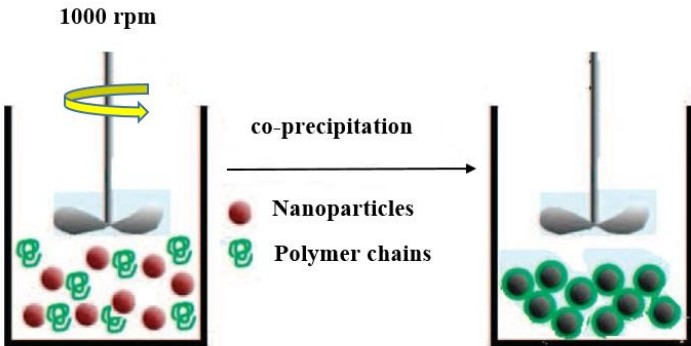

**Figure 18.** Schematic illustration of solution blending method.

### 5.4. Melt Blending

Melt blending necessities the direct dispersion of nanofillers into the molten polymer. When the mixing process starts in its melt state, the resulting polymer strain that is applied on the particles is dependent on the weight distribution and the weight of the molecules. The size of the agglomerates decreases when the shear stress level is high [157]. At the beginning, the larger agglomerates break apart to become smaller in size before being dispersed within the polymer matrix. Stronger shearing results when the polymer strain is transferred to the new agglomerates. Individual particles are formed due to the breaking down. The primary element of this method is the timing and the chemical processes between the NPs surface and the polymer [162].

Melt blending necessitates single and twin-screw extruders [163]. However, there are occasions where unfavourable outcome may ensue on the nanofiller's modified surface due to high temperatures thus optimisation is applied to address this issue [164]. The most renowned method to address this is to use intermeshing co-rotating twin-screw extruders. The disadvantage of such method is the difficulty in controlling the parameters such as interaction between the NPs, polymer and the procession conditions such as residence time and temperature [165]. As such, it is not easy to achieve NPs that are evenly dispersed. Melt blending can be commercialised as it is compatible with a range of industrial operations including extrusion and injection moulding [157]. The main advantages of this are low cost, environmentally sustainable due to the absence of solvents, heat stability enhancement [166], improved mechanical properties [167] and good NPs dispersion [168]. Its disadvantage is the possibility of damage on the nanofillers' modified surface as a result of the high temperature application [169]. Overall, each method has its own respective advantages and disadvantages and the selection should be based on the conditions and underlying materials.

### 5.5. Nanopolymers and Their Applications

Nanopolymers offer rife of applications including all the uses offered by the traditional polymers. These applications include telecommunications, defence, household goods, daily services, utilities and basic utilities, etc. Further details include the plastic containers, toothpaste, and so forth. Nanopolymers are favoured due to their many notable attributes included high resistance to chemical, excellent tensile strength capacity to hold metals and other compounds. High conductive properties of nanopolymers enable their usage in nano circuit fabrication. There is possibility to produce polymer with nanoparticles from many different structures where some can be self-assembled such as lamellar, lamellar-within-cylindrical, lamellar-within-spherical, spherical-within-lamellar and cylindrical-within-lamellar geometry. The examples of non-self-assembled structures are polymeric nanocapsules, polymer brushes, nanofibers, hyperbranched polymers, dendrimers and polymeric nanotubes.

There is still a constant renovation in the nanotechnology due to the great demand for practical applications. Nanofibers made via electrospinning find many uses within the environment. Due to their remarkable length and ability to embed in other media,

nanofibers became one of the safest NMs. Other desirable properties include high porosities (over 80%), adjustable functionality and high surface-to-volume ratio. These characteristics are more effective than the conventional non-woven and polymeric membranes especially those use in the liquid filtration and particulate separation. It is feasible to apply the nanofibrous scaffolds exclusively as a cutting-edge component for the liquid separation and gas filtration. Due to the advancement of electroblowing and electrospinning technology high performance nanofibrous scaffolds became feasible. In this perception, it is customary to highlight various applications of nanofibers for the solar energy harnessing and as membranes to remove heavy ions from the industrial wastewater and discharges.

## 6. Nanotechnology Based Smart Glass Materials

The conventional usage of high-performance glazing systems is upon windows or building windows in order to decrease the amount of unwanted heat from the sun as well as reducing the workload to cool air from the air condition systems installed within the building. Aesthetically, glass facades are more desirable in terms of using them in commercial buildings [170–172]. Thus, careful studies are needed to estimate and evaluate the energy savings in practical terms for the high-performing glass. It is particularly vital for the research to be conducted on the different types of high-performance building glass. This is especially true for densely built cities such as Singapore where the heat from the sun is an issue for the buildings. Additionally, it is still uncertain whether these glasses are able to retain its efficiency in countries with four seasons. The glass's U-value needs to be within acceptable range in order for it to be functional when subjected to various climates including the tropics.

Assessment of glass performance necessitates active measurement while being subjected to a controlled source of radiant. Evidently, such testing environment may not account for actual weather, where many possibilities may not be duplicated within the test environment [171,173]. This may be compensated by subjecting the glasses to real weather conditions by actually installing them outdoor [174,175]. However, it is still not possible to conduct testing in large-scare and fast on-site characterization. Furthermore, the test is restricted to fabricate the glazing, indicating the impossibility to forecast possible problems during the design stage. The development of cutting-edge technique made it possible for professionals to be able to make simulation and assessment on the glass being installed in building during the design stage [174,176,177]. Over the counter and matured simulation codes that are open source in nature including Energy Plus and Radiance [174] are appropriate for such assessment as they have undergone development spanning for more than ten years. Despite such tools, the assessment is still complex in terms of conducting such evaluation for high performance glazing description (glass with various coatings for many purposes) into the solar irradiance module through the use of the current glass models.

The glass involving multiple layer or pane glazing must be classified and computed uniquely using more focused tool prior to be interfaced using a custom script. Further consideration is vital in terms of acquiring complete and comprehensive weather model to increase the accuracy of the solar heat gain that will enter into the building. With the exception of weather data from International Weather for Energy Calculations (IWEC), other data must be carefully considered when inputted for assessment using the aforementioned tools. In addition, the tools are developed with main consideration to indoor performance thus it does not account for the negative effects and other impacts of the sunlight being reflected off the building glass façade during the assessment for environmental risks. External use of the glass typically involves alternative glass material, which are mirror or opaque with high specularity. The use of such alternative materials means that their properties have no angle dependence [174].

Usually, the glass performance is evaluated via the active measurements by utilizing the known radiant sources. While, under the non-controlled weather there is no possibility to apply this type of setup. Though a passive measurement can be running under real weather conditions using the outdoor test chamber, however it is unsuitable for a large-

scale testing and fast on-site characterization. Additionally, the mentioned test is limited to the fabricated glazing and thus unable to predict potential issues in the design stage. Advances in the simulation techniques have enabled the building professionals to evaluate the glass facade of a building at the design phase. Nonetheless, the typical simulation tools are unable to integrate the high performance glazing description. Using the advanced coating technology, although the existing glass models can be tested but these tools often lack local weather models that plays an important role in accurately assess the solar heat gain by the building. Nanotechnology is being applied in various disciplines especially within construction materials due to its ability to decrease the consumption of energy thus they have much potential. Glass is one of the most special construction materials and can be treated with nanotechnology, decreasing the transfer of heat through the building envelope (Figure 19). The study used Design Builder 3.1 and followed the Egyptian energy code requirement to assess the difference energy consumption between two types of glass, standard 6 mm clear glass and glass that is treated with nanotechnology. The standard 6 mm clear glass that were used in glazed facades results in high thermal loads into the indoor environment of the building. This results in increasing use of energy in the building [178].

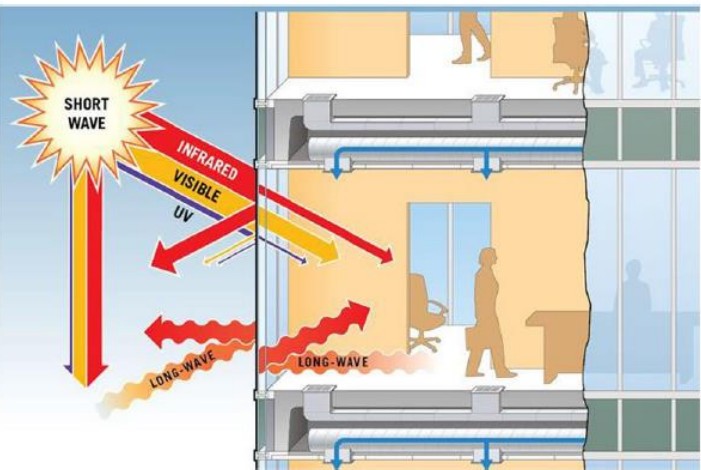

**Figure 19.** The glass treated with nanotechnology [178].

Glass is a common material within various industries including transport, building and construction, solar energy with glass variety. It is also being used in microscopes, tablet computers, furniture and many more. There are four advantages of using glass within the building and construction sectors. Firstly, it allows natural lights to enter the building. Secondly, it filters out harmful rays from the sun from entering the building. Thirdly, it harmonises the environment and the building. Lastly, it is cost-effective due to its energy efficiency. Researchers and scientists have taken the motto of 'necessity is the mother of invention'. Thus, during their research in improving the properties of glass, they have developed a type of glass that requires minimal maintenance, also known as self-cleaning glass. Individuals that wear glasses will be glad that such glasses prevent mist from forming when they are enjoying hot drinks that are steaming or when they are cooking. In addition, anti-fogging glass is used in tablet computers thus they could be used in close proximity to swimming pools. In addition, anti-reflective glass is used in mobile phones or laptops thus users could still able to use these devices during broad daylight. As for self-cleaning glass, they are most suitable for windows and doors in offices and homes, where these SCGs do not need any frequent maintenance for cleaning.

*6.1. Self-Cleaning Glass*

Glass is extensively used by diverse industries like automotive, solar cells, building and construction. SCG is a new type of glass being developed and is widely used in hard-to-reach areas in buildings because it requires minimal maintenance. SCG has either

a layer of titania ($TiO_2$) that measures 10–25 nm or is coated with silica on its surface via both bottom-up or top-down approach. The self-cleaning properties are the control of its wettability properties on its surface. The first is for the surface to be complete dry, also known as hydrophobic surface, where a liquid droplet maintains a spherical shape on the surface of the glass. This is achieved either by forming a component of low surface energy or through surface roughness control. The surface becomes hydrophobic by applying a thin layer of $SiO_2$. The second technique involves complete wetting of the surface where the liquid forms a film upon contact with the surface. This is known as hydrophilic and can be achieved through applying photocatalytic $TiO_2$ coating. The coating uses the sunlight and water to rinse itself thus resulting in self-cleaning property. Therefore, solid surfaces undergo various reactions with dissimilar materials depending on the coating type being applied. Consideration should take place upon the various qualities as a result such as spreading, wettability, adhesion and interface. The wettability property of a solid is defined by observing that contact angle (denoted by θ) the moment liquid touches the surface of the solid.

### 6.2. Hydrophilic Coating

A surface is deemed hydrophilic when the water contact angle (CA) is less than 90°. It is considered as super hydrophilic when its CA is less than 50°. As the liquid contacts such surfaces, it will spread out until it becomes a thin layer. The self-cleaning materials that made this possible are $WO_3$, $ZnO$, $SnO_2$, $SiO_2$, $CdS$, $TiO_2$ and $ZrO_2$. The most extensively used is $TiO_2$ because it has more advantages in comparison with the others. New discovery made by Fujishima and Honda where they used $TiO_2$ for photo-electrochemical splitting of water to hydrogen and oxygen while being subjected to UV radiation. This has resulted in an explosion of research to study the $TiO_2$ photo-catalytic potential including self-cleaning coatings, photo-electro-catalysis, photovoltaics, photoelectrocatalytic degradation of organic compounds and advanced oxidation. $TiO_2$ exhibits the following properties, high refractive index, good mechanical performance, transparent and semiconductor material with a high band gap. When $TiO_2$ is within the wavelength range from 0.35 mm to 12 mm, it becomes stable chemically. Titania exists in three different crystal structures such as brookite, anatase and rutile. The highest refractive index is shown by the rutile phase (2.61–2.90), making it the centre of focus for optical applications. Rutile is also the most stable in terms of its thermodynamic properties especially when subjected to high temperatures. Despite various advantages, anatase has increased desire for lower temperature applications where it is necessary to form a film on thermally sensitive substrates. Therefore, the desirable materials are amorphous or crystalline anatase, used to produce the self-cleaning glasses at temperatures below 400 °C. Anatase can be changed to rutile in the range of 700 to 1100 °C.

### 6.3. Anti-Reflective Coating

Fujishima and Guiselin et al. invented the $TiO_2$ thin films and also patented the methods. This film being transparent, photo-catalytically efficient and abrasion-resistant can be used on glass surfaces. Several SCG are already being commercially used at the present time such as Hydrotecht from TOTO, Activt from Pilkington Glass, Thermotecht from Viridian and Bioclean from Saint Gobain. Additionally, self-glazing products are also being rolled out in liquid forms or white that target direct consumers. When a normal glass is applied the self-cleaning products, they would turn into SCG. Products that are available for users are produced by some companies such as Rain Racert from Rain Racer Developments, BalcoNanot from Balcony Systems Solutions and ClearShieldt from Ritec International. SCG can be installed in various locations including offices, facades and general buildings. The improvement of photo-catalytic activity and anatase coating necessitate a high refractive index due to low temperature processing.

Other properties, apart from self-cleaning, are necessary for a glass that will be used on smart phones, spectacles and solar cells. These properties include anti-fogging, anti-abrasive and anti-reflection. Fraunhofer is the founder of anti-reflective (AR) coating in 1817.

Since then, AR phenomenon is regarded as a destructive interference between air-coating interfaces by Fresnel and Poisson and light reflected due to substrate coating. One of the many methods of making AR coating is to construct a single-layer of coating that has low refractive index. Materials that have low refractive index cost more and also rare. Porous nanostructures can be used to effectively decrease the volume-averaged refractive indices of materials through controlling the porosity within the coatings. This results in anti-reflective coatings, and its hydrophobic and hydrophilic properties could be further enhanced by increasing surface roughness. At the same time, reflection is increased due to the decreased in transmittance, which occurs as a result of scattering diffusion in rough surfaces. Sample transmittance is the subsequent light intensity ratio that exits the after intensity ratio entered the sample. Therefore, the increase of transmittance results in decrease of photocatalytic activity as the light intensity decreases. At the same time, anti-reflective surfaces are part of the SCG. Therefore, in order to preserve the self-cleaning and anti-reflectivity properties, the ideal surface roughness is required. The assessment of the solid surface's wettability necessitates the static contact angle and the dynamic sliding angle. The essential factor is therefore roughness of the surface and chemical functionalization.

### 6.4. Fabrication of Self-Cleaning Glass

The glass becomes either hydrophilic or hydrophobic after the applications of a thin layer of $TiO_2$ or $SiO_2$ on its surface. There are two types of fabrication of nanomaterials which are top-down and bottom-up. The top-down approach involves removal of materials gradually from massive structure until the required nanomaterial is formed. Lithography is an example of this method. Comparatively speaking, it is similar to using a block of wood and turning it into a doll by a carpenter. Bottom-up approach involves the use of atoms or molecules to be built gradually until the formation of the required nanomaterial or nanocoating. Comparatively, this is akin to using Lego blocks to build a house. The bottom-up approach is further divided into two types like gas and liquid phase. The gas phase method involves the plasma arc evaporation and chemical vapour deposition (CVD). The liquid phase technique deals with the sol-gel and molecular self-assembly.

### 6.5. $SiO_2$-$TiO_2$ Coating

In addition to having self-cleaning function, other functions are also desirable including photocatalysis and anti-reflectivity, which are vital in products such as smart phones and solar cells. Glop et al. [179] used the sol-gel method and Liu et al. [180] used the pulse magnetron sputtering as methods of preparation for the photoactive antireflection coating. The $TiO_2$ coating on the outer surface that results in self-cleaning feature increases the reflectance of plastic or glass substrate due to its relatively high refractive index (c. 2.5 for the anatase phase). Therefore, self-cleaning and anti-reflectivity attributes may not be compatible with the exception of rare instance where the structure and composition are modulated. Prado et al. [181] attempted to produce a coating that is multifunctional where its outer layer consists of dense/mesoporous $TiO_2$ and its inner layer consist of mesoporous $SiO_2$ AR layer. Multifunction coatings with self-cleaning attribute have discovered to perform 25–30% compared to photo degradation degree, which is produced via the conventional $TiO_2$ coatings layer either porous or compact. Solar industry including solar power plants and solar energy producers primarily use glass. The amount of electricity generated or power for heating water depending on the intensity of the sunlight. The glass may be useful in terms of reducing loss of radiation and reflection.

SCG's primary property is its reflective index denoted by n. Production of glasses with anti-reflective properties for solar related use requires a low refractive index such as $SiO_2$ where its n value is 1.4. Conversely, high reflective index such as titania where its n value is 2.0 is vital in improving the photocatalytic activity of the hydrophilic property in SCG. Helsch and Deubener [182] attempted to use the sol-gel coating technique to create a single type of glass that contains both functions. High transmittance is needed for this particular type of glass. The research has been a success where they used two layers consisting of $SiO_2$

and $TiO_2$ to create a glass with both anti-reflective and photocatalytic properties. Through the sol-gel coating method, preparation was made on silica glass porous coatings $xTiO_2$. $(1002x) SiO_2$ with 50 wt% of titania. The compatibility of anti-reflective and photocatalytic properties will then be achieved once the composition reached the ranger from x57.5_20. Porous coating also enhances the solar transmittance by 2.3% in comparison to silica glass that is not coated. These coatings with dual functions have greater degradation rate at 20-fold of the air borne contaminants in comparison to nanoporous film of pure $SiO_2$.

Nanoporous structures consist of materials that have high porosity and low density with simultaneous advantages in terms of possessing high pore volume, high surface area and larger pore size, whereby the diffusion pathways are accessible. Anti-reflection coating is often made of porous silica layers. Helsch et al. [183] made a discovery that there is a 5% enhancement (from 92 to 97%) of light transmission when borosilicate glass is at 550 nm, 35% porosity and 110 nm film thickness. There are many applications for $TiO_2$ films on glass substrates including mirrors, windshields and window glasses. While being serviced, anti-reflective porous coatings will be subjected to severe environmental conditions including hail and salt atmosphere, sandstorms, dust particles and airborne volatile organic compounds. If the AR coating is damaged while being subjected to the aforementioned conditions, the solar transmittance will be reduced. Cathro et al. [184] discovered an increase on the refraction index of porous thin films as a result of the adsorption of airborne contaminants. Pareek et al. [185] found that oil vapour contamination is also responsible for the increase of the refractive index of porous antireflective coatings.

*6.6. Nanomaterial-Based Solar Cool Coatings*

The global building and construction industry is responsible for both 40% of the entire world's energy consumption and emitting a third of the world's greenhouse gases annually. At least 50% of the total energy consumed by this industry are for powering heating, ventilating and air conditioning (HVAC) systems. Passive cooling and solar heat insulation technologies are often being regarded as solutions in addressing the global energy crisis, in which they are being considered as reducing or even consume zero energy. Solar radiation plays the main role in terms of buildings gaining heat when they are transmitted via the envelope. The heat will be trapped and increased inside the building. Buildings therefore, are more likely to use nanomaterials-based solar cool coatings (NSCCs) in order to address the issue of excessive solar heat and energy consumption. These coatings are currently the most reliable in terms of passive cooling technologies. NSCCs are composite materials where it is made of thin-layered substrates mixed with nanosized additives, which is the primary component due to its distribution solar reduction function onto a normal coating material. Binders are made of thin-layered substrates and they are added with nanosized additives to provide a coating to the surfaces of buildings where required.

NSCCs are widely used for the past few years as a solution to the high energy consumption in buildings. On [186], authors have conducted a market research to show that the solar coatings will have a 70% in saving energy on a global scale from 2013 to 2019. The number of patents being filed is evidence to the increased attention and research being taken place to increase the use of NSCCs as well as proving that it is a pioneering technology in passive cooling. On [187], authors have revealed that there is a significant increase by 38% between 2013 to 2015 on the number of patents being filed on smart window coatings. Meanwhile, the patents for thermal barrier coatings have increased by 32% between 2011 and 2015. For several decades, the key component in transparent coatings used in solar heat reflection is metal. Al, Au, Ag, Cu and Pt have higher performance in terms of possessing high reflectivity and low absorptivity properties. Incorporating these metals into NSCCs leads to reflection of solar heat that would have otherwise penetrated into the indoor environment of the building. This passive cooling feature means that the indoor environment will need to use less energy consumption for cooling purposes. Significant amount of research was conducted on solar cool coatings. The least difficult in terms of implementation and usage are Au and Ag. Cher (2014) prepared nanogold (Au) films to be applied on the glass

surface using aerosol-assisted CVD method wherein the deposition was performed inside a cold-walled horizontal-bed CVD reactor. The resulting nanogold layers possessed various morphologies depending on different reaction temperatures. Observation was made by placing a layer of Au NPs at 500 °C (Figure 20a). Figure 20b,c shows individual Au NPs on the top plate of films subjected to 400 °C. The results disclosed that the thermophoresis was responsible for the increase of particle size, wherein the NPs formation and gold atoms aggregation occurred in the gas phase reactions prior to the deposition.

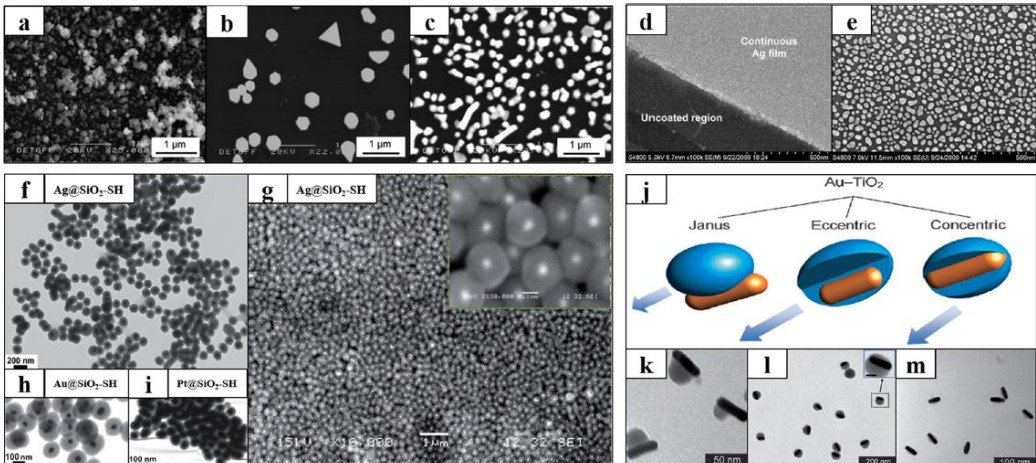

**Figure 20.** (**a**–**c**) SEM (Scanning Electron Microscope) images of nano-Au deposited on the top plates at various temperatures. (**d**,**e**) SEM images of an Ag layer (**d**) with and (**e**) without a Ge wetting layer. (**f**–**i**) TEM (Transmission Electron Microscope) and SEM images of Ag@SiO$_2$-SH (**f**,**g**), (**h**) Au@SiO$_2$-SH, (**i**) Pt@SiO$_2$-SH. (**j**–**m**) Schematic (**j**) of Au@TiO$_2$ nanorods with various geometries; TEM images of Au@TiO$_2$ nanorods with (**k**) Janus, (**l**) eccentric, and (**m**) concentric geometries [187]. Reproduced with permission from Zheng, L., et al., Solar Energy; published by Elsevier, 2019.

## 7. Environmental Health and Safety Considerations

The influence of nanotechnology is apparent in industry and many aspects of life; construction is no exception. Even though enhanced-quality materials equipped with innovative features are already being used, numerous potential applications of nanomaterials still exist in the field of construction that are yet to be capitalised upon. However, these endeavours do not come without risks. Negative outcomes and effects on the environment and human health are not outside the realm of possibility. Hence a prudent and cautious approach should be considered. There are several existing nanoparticles, such as titanium dioxide and carbon nanotubes, that could already be harmful to those individuals tasked with their direct use. Qualitative and quantitative risk assessments, occupational health and safety risk management, and adequate circumvention protocols for identified risks are not only important but are crucial to avoiding or mitigating potential disaster.

Nanomaterials are so many in number and so varied, it is safe to assume that massive quantities of these materials will eventually be produced. Moreover, introduction of entirely new nanomaterials both trigger the requirement of adequate risk assessment procedures and suitable communication measures surrounding those risks. Presently, new nanomaterials are analysed in a manner similar to that used for chemicals, food, and consumer products, which is unsurprisingly both inefficient and insufficient. The challenges presented when characterising nanomaterials and generating a standardised processing approach become substantial bottlenecks to the process. However, there do exist several techniques; laser ablation inductively-coupled plasma mass spectrometry—that could stand to meet the needs of such processes. For example, aiding in the quantification of nanomaterials as subsets of complex matrices.

Despite existing awareness surrounding the potential risks for working with construction nanomaterials, and the notion that these materials may even pose risks to end-users,

hazard information remains limited [188]. Consequently, the Occupational Safety and Health Administration (OSHA) has no recourse to mitigate the unknown hazards of Nutritional risk screening (NRs), as it is without regulations nor enforceable exposure limits; this is regardless of the fact that nanomaterial contamination can take place at any time during the manufacturing, packaging, and transport of construction materials, their use on-site during construction, and after the work is complete during the operational phase. For example, a number of workers were shown to have been exposed to more than the recommended limit of titanium dioxide during the packaging process in a study conducted by Al-Bayati and Al-Zubaidi [189]. In a recent move to promote safe working practices, the CPWR developed a toolbox talk strongly recommending and endorsing the use of high efficiency particulate air (HEPA) filters when handling nanomaterials. This was in response to the discovery that construction nanomaterials can be converted into unintended forms when mass-manufactured [190], such as carbon-based nanomaterials becoming airborne when prepared as a solution. However, it is worth noting that HEPA filters were never designed to capture particles of under 300 nm in size, making it unlikely to eliminate the hazard, even though they may still serve to mitigate it.

As mentioned before, a significant impact stands to be made by the use of nanotechnology within the construction industry, not only from a perspective of enhancing material properties, but also because a high proportion of all energy used by the world is consumed by commercial and residential buildings, in their lighting, heating, and air conditioning. Overtaken thus far by the adoption of nanotechnology within fields such as biomedical and electronics, the construction industry has been making up lost ground in their pursuit of innovation using a variety of nanomaterials in recent years. However, as alluded to previously, adoption of novel technologies does not come without risks; the potential dangers to the environment and human health posed by nanomaterials should not go unconsidered. This is true even if the goal in their use is to preserve the environment, by utilising the energy-conserving functions provided by nanomaterials, their full lifespan must still be contemplated, as highlighted in a recent review by Rice University scientists. Unintended consequences could be far severe than those it was intended to prevent. Furthermore, the authors indicate that nanomaterials, especially CNTs, can be accidentally or incidentally introduced to the environment at various stages of their life cycle.

Within their work at Rice, they go on to detail the importance of a holistic nanomaterials' lifecycle exposure profiling approach, stipulating without that level of meticulousness, critical impacts on ecosystem and human health cannot be avoided. They maintain that, as a result of no regulation being presently in place despite growing concerns, a number of MNMs should be regarded as 'potential emerging pollutants' until contradicting information surfaces, as there are many related risks to environmental and public health that are being disregarded without that regulation. Furthermore, they describe the element of unpredictability of the natural environment; once distributed into it, nanomaterials may transform in diverse chemical, biological, and physical fashions, altering their properties, effects, and ultimate fate.

The potential routes along which nanomaterials can be released into the environment are many and often. From occupational exposure, when the material is first being prepared, during any coating, moulding, incorporating, or compounding to contamination during installation, construction, maintenance, repair, renovation. Finally, to decommissioning or demolition processes, even beyond this stage, further risks arise when solid nanomaterials reach landfills or get disposed of in incinerators. Delivery methods and approaches affect these risks, also: aerosolization of nanomaterials, adhesive wear, abrasion and corrosion, and manufacturing process wastewater effluent outlets all have additional risks, specific to the method and altering the resultant hazard.

## 8. Using Nanomaterials Safely

The question "how to utilise nanomaterials safely" does not find itself wholly resolved, even though it is clear that discovering it is crucial to improving the performance of infras-

tructure and buildings. These nanoscale fibres and particles could already be contributing to a problem that the scientific community is as yet completely unaware of, or in the ways of which we know that they can. Thin strands carried airborne can acquire behaviour patterns akin to asbestos. Limited information is available for workers and manufacturers alike on keeping safe while handling these materials, while it is commonplace to appreciate the necessity of greater regulation.

Given that estimates place up to half of all new building materials in 2025 as containing nanomaterials, this information is urgently sought. This was the motivation for the research team at Loughborough University, when they investigated where these materials are used, to what extent, number of potential risks, and how might the workers on the 'front line' mitigate these risks. It was funded in part by the Institution of Occupational Safety and Health (IOSH), in order to produce a framework and a measure of guidance.

An additional challenge facing the generation of a set of guidelines as such, or indeed, any other form of regulation, is that the way health and safety legislation is applied in different countries. It may not be mandatory for manufacturers to specify information about the type of nanomaterial, or the approach with which it was used, resulting in largely unreliable and inconsistent labelling systems.

## 9. Conclusions

The construction industry has witnessed an ever-increasing applications of various sustainable materials using the core-shell strategy and nanotechnologies. The following conclusions are made based on the in-depth and relevant literature overview of nanotechnology-based core-shell structures:

i.  A new class of hybrid and core-shell NPs can be developed due to the advent of the manipulation techniques of particle structures at the nanoscale.

ii.  Efficient fabrication methods are now available for the large scale production of numerous types of core-shell nanostructures. These developed techniques have contributed to the fast-paced advancement of synthetic chemistry, device setup, colloid and interfacial science.

iii.  The pigments durability can remarkably be improved using the core-shell NPs. Furthermore, being a part of sustainable materials, these NPs have widespread applications. The highest recommended materials for shells in the construction industries are $SiO_2$ and $TiO_2$.

iv.  Carbon-based nano-enhancers show higher thermal conductivity compared to metals or oxide-based materials. High surface affinity between the organic structures and carbon nano-fillers of PCM can enhance the uniform interpenetration and lower the particles' scatterings at the interfacial surfaces.

v.  In the near future, the high-performance nano-enhanced phase change material technology will be of great demand. It is expected to be applicable in many areas particularly in the thermal storage within the sustainable and renewable energy field. These applications include the solar energy power generation, industrial heat charging/discharging processes, excess heat management and cooling of electronic devices.

vi.  The polymer nanocomposites have immense applications potential compared to the traditional materials. Thus, nanocomposites field has been the popular research topic due to its several desirable features including ease of production, light weight and flexibility. The most distinguishing aspect of polymer nanocomposites is their utility small fillers, resulting in a significant increase in the interfacial interactions than the conventional composites.

vii.  It is foreseeable that the core-shell NPs will continue to play a significant role in the passive cooling technology, reducing the solar heat gain by the buildings and energy consumption. In the context of global climate change and fast urbanization-mediated energy deficiency and environmental deterioration, the development of core-shell NPs is expected to be faster mainly in two aspects like the large scale synthesis of

high performance nanomaterials and cost-effective as well as time-efficient coating fabrication techniques.

viii. Accompanied by the standardised approaches and regulatory mandates, high-volume nanomaterials such as $SiO_2$ and carbon black, play a major role for a variety of industrial applications. Despite still being in development phases for many applications, the appropriate analytical capacity for the characterisation of materials and their properties are still necessary and fundamental; existing reports from toxicological inhalation studies already indicate steady increases of nanomaterial toxicity, as opposed to demonstrating entirely new nano-specific effects and outcomes. These effects are improbable, however, as a result of the rather arbitrary and flighty nature of the definition of 'nanomaterial'. Nevertheless, while nano-dimensions themselves do not present toxicological hazards, the sheer quantity of new materials such as hybrids and composites make further attention and study mandatory.

**Funding:** This research received no external funding.

**Data Availability Statement:** Not applicable.

**Acknowledgments:** Authors thank National University of Singapore for their support and cooperation to conduct this research.

**Conflicts of Interest:** The authors declare no conflict of interest.

## Abbreviations

| | |
|---|---|
| NPs | Nanoparticles |
| CNTs | Carbon nanotube |
| SWCNTs | Single-walled carbon nanotube |
| TMAOH | Tetra methyl ammonium hydroxide |
| EG | Ethylene glycol |
| SEM | Scanning electronic microscopy |
| XRD | X-ray diffraction |
| TEM | Transmission electron microscopy |
| PCMs | Phase change materials |
| NMs | Nanomaterials |
| PNCs | polymer nanocomposites |
| HEPA | High efficiency particulate air |

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
