# Peer review of "Potential Applications of Core-Shell Nanoparticles in Construction Industry Revisited"

_2673-3501, doi:10.3390/applnano4020006_

Round 1

Reviewer 1 Report

Comments and Suggestions for Authors

Ghasan Fahim Huseien presents a very interesting and complete presentation of the most notable core-shell materials, their fabrication, characterization and application. The work reads nicely and is interseting and I have just some minor comments that however should be adressed prior to publication:

1) Sometimes the use of the English could be improved. The author should cross check.

2) The figure numbering is sometimes confusing e.g. Figure 1.8 followed by Figure 9.

3) The author should ensure that all permissions for reprint have been collected for the figures.

4) The author often uses abbreviations that are not explained, in particular when describing the samples of cited articles (e.g. : CT, CFT2, CFT4, CFT5, and CF. CT). The author should be more clear about this and explain what such abbreviations stand for and what material the sample is.

Author Response

Reviewer' comments are highly appreciated. Kindly find the attached file (author response).

Reviewer 2 Report

Comments and Suggestions for Authors

The review paper entitled "A Review on Applications of Core-Shell Nanoparticles in Construction Industry" focuses on the core-shell nanoparticles for the applications in construction industry such as pigments, self-cleaning process, phase change materials, polymer composite, and building glass enhancement. The effects of the samples’ core-shell type, size, shape, and content for the relating properties and applications are comprehensively discussed. However, the title is focused on the “core-shell nanoparticle”, so the contents of this paper should intensively relate to this structure. But the cases for the applications in such as phase change materials or polymer composite sections is not so related to this structure that is defined on pages 5; so, it is recommended that author should provide more cases in these systems.

Specific comments:

1.     More description relating to the important of the core-shell nanoparticle materials should be added. What’s effects will introduce into the materials with core-shell structure. For example, the difference and improvement in optical properties so that the core-shell structure can be used for pigment applications.

2.     More cited papers in recent 3 years should be added.

3.     The organization of this paper should be adjusted. It is hard to catch the important point for the core-shell nanoparticle and the applications in construction industry in recent content.

4.     It is recommended to add a table relating to the advantages and disadvantages for the core-shell nanoparticle structures.

5.     Some number of Figures should be corrected, such as the figure 1.7 and 1.8, or the description on page 11, line 478-479.

Author Response

Kindly find the attach file. 

Reviewer 3 Report

Comments and Suggestions for Authors

The manuscript can be appected in present form

Author Response

Reviewer' comments are greatly appreciated. Kindly find the attach file. 

Reviewer 4 Report

Comments and Suggestions for Authors

Comments for applnano-2146743.

The review paper is quite comprehensive and interesting, summarized a vast range of recent advances in syntheses and development of core-shell structured NPs used in construction industry. However, there are some minor suggestions that could improve the quality of the review paper.

Suggestions:

Firstly, although the materials are well written and comprehensive but that’s make it excessive and way lengthened. If it is condensed, it will make it more interesting. Secondly, if the author put a table which list different core shell nanoparticles (presented in figure 2) and the method followed for their synthesis along with their application will make it concise but panoramic.

Minors:

Line 80-82; rewrite the sentence.

Line 149, 150; Figure 1.2 could be a typo.

Line 160,161. Remove the space between “chemical / physical”.

Line 226-228. “synthesized” Rephrase the sentences.

Line 252. Should be “Tetraethyl orthosilicate (TEOS)” use of full form and abbreviated form at first appearance and then only abbreviation in proceeding texts.

Line 265. Figure 3 and others, legend should mention the copyright statement of the images reproduced.

Line 270. “before using it coat ZnO NPs” rephrase it.

Line 277,278. Rephrase the sentence for clarity.

Line 303. “ethylene glycol (EG)” use of full form and abbreviated form at first appearance and then only abbreviation in proceeding texts. Appeared already in Line 280.

Lines 336, and 349; should use the author’s names before [46], [47].

Line 502, Liu et al’s. correct it.

Line 524, 539, Figure 1.7, Figure 1.8 seems to be figure 7 and figure 8. Please have a look and confirm it.

Line 543, On [86],?

Line 567, Figure 1.10?

Line 612, a sentence shouldn’t start with a digit.

Line 1111, the space between heading paragraph is inconsistent.

Line 1140, layer / pane, remove the space before and after slash symbol.

Line 1234, delete the space before despite.

Line 1315, SiO2 / TiO2, remove the space before and after slash symbol.

Line 1410, v. vi.In the foreseeable future, Conclusion point v, vi should be corrected.

Line 1438-1440, follow journal guidelines.

Author Response

Reviewer' comments are highly appreciated. Kindly find the attached file. 

Reviewer 5 Report

Comments and Suggestions for Authors

The Authors wrote an extensive review paper that covers topics of core-shell nanoparticles, phase changing materials, polymers and composites, and smart glass. All these have possible use in construction in common. However, the paper is badly structured and such looses all the value the authors tried to bring to the readers. At this point, the Reviewer would recommend rejection of the paper. Please, read the arguments bellow. Furthermore, the Reviewer gives some suggestions how to rewrite the paper, before resubmission.

The Introduction is inadequate;
Authors try to describe what nanotechnology is, and what influence in the World it could have, both, positive and negative. In addition, authors open one or two philosophical topics, which might be, or not, interesting. Unfortunately, the text is not concise, nor any idea properly developed. Furthermore, the introduction only vaguely touches the topics of "nanotechnology in construction" and "core-shell nanoparticles", which are suggested by the title. Another major issue is, that the more than half of the paper don't have the topic of core-shell nanoparticles; after the chapter 3, topic of the paper switches from the core-shell particles and the next time the phrase "core-shell" is mentioned is in the chapter 7 - Conclusions. The topics, covered in the paper, are i) core-shell nanoparticles; ii) pigments; iii) phase changing materials; iv) polymers and composites; v) smart glass materials.

The introduction should be rewritten from scratch to enlighten all the topic of the paper, give an idea of the structure of the paper and what is the papers purpose.

It seems all the topics covered in the paper could be used in construction, and all can involve nanotechnology. The title for the paper should thus be changed appropriately. In addition, index for the paper is suggested.

The body chapters:

Chapter 2; Core-shell nanoparticle: Synthesis approach and importance
Due to the pore writing it seems as if authors wouldn't understand the processes of synthesis of core-shell nanoparticles; the information is given chaotically and there is no added value for the reader. Furthermore, this paper couldn't serve as a reference, a starting point for further education since a lot of crucial information is missing, as well as proper citations.

Reviewer would suggest to rewrite the chapter in a way, to, first: make a short introduction, why the core-shell nanoparticles should be used in construction, which problems they solve, or which problems they have at the currently known solutions. Second, list the techniques that are able to synthesise core-shell nanoparticles, describe the techniques and state what is possible to obtain by them. Then, comparison between the techniques should be made: what are the advantages and disadvantages of each technique. At the end, the most promising technique for the core-shell nanoparticle synthesis should be exposed by the criteria of construction, if possible. That would give the value to the paper and the interested reader.

Chapter 3; Nanomaterials based sustainable pigments
Why are the authors describing pigments? The purpose should be stated in the Introduction chapter already. Furthermore, in the chapter itself, the importance of pigments is introduced by sentence: "Recent research has shown that efficient energy consumption and environmental protection measures are deemed significant [52]." But what is the connection between energy consumption and the pigments? That would be a proper introduction to the chapter, possibly also in the chapter Introduction as well. All the other comments for the Chapter 2 apply here as well.

Similar trends continue for the other chapters, and same comments can be applied. The Authors also don't do a critical review of the topics in the chapters, which leads to the fact, that no substantial conclusion are made. This results in too general conclusions in the Chapter Conclusions, which are not clearly discussed in the paper.

The Authors touch the topic of nano-safety and nanomaterial environmental influence. The Reviewer agrees that these topics are extremely important, especially, if the nanomaterial is to be widely used in construction. If Authors can give some guidance, then another chapter devoted to this might be suggested. But probably just a better delivered warning with some citation in the Introduction would suffice, as the topic could cover another paper.

Last, text should be simplified. The reader gets a feeling that the Authors want to use complicated English, but the result is hard to read, plenty of sentences are grammatically incorrect, and sometimes the Authors contradict the point they were building by mistake. Furthermore, the amount of text should be drastically reduced.

Author Response

(The authors gave the same response as above.)

Round 2

Reviewer 2 Report

Comments and Suggestions for Authors

The manuscript has been revised as per earlier comments by reviewers.

Author Response

Reviewer' comment is highly appreciated. Please see the attached file. 

Reviewer 5 Report

Comments and Suggestions for Authors

Dear Authors and Editor,

Although the Authors have selected relevant literature, presented interesting cases, and included valuable information in the paper, the language and presentation of the manuscript still require improvement. The text contains vague statements, loose claims, and some sentences without a clear meaning. The Introduction section lacks clear direction, relevant information, and an introduction to the rest of the paper. Furthermore, the manuscript is still too long, making it challenging for readers to follow the paper's structure and understand the information presented. Despite my previous recommendations, none of them were properly taken into account. Therefore, I regret to inform you that my recommendation is still to reject the paper in its current form.

I hope that the Authors will consider the feedback provided and revise the manuscript. With the necessary improvements, the paper has the potential to reach a broad readership and have a significant impact in the field. The Authors are encouraged to restructure the paper, shorten it, and correct the English language. I hope that this feedback will be taken constructively, and the manuscript will be resubmitted after revision.

Author Response

Reviewer' comments are greatly appreciated. Kindly find the attached file. 
